# Highly reversible zinc metal anode enabled by strong Brønsted acid and hydrophobic interfacial chemistry

Qingshun Nian [1], Xuan Luo[1], Digen Ruan[1], Yecheng Li[1], Bing-Qing Xiong[1], Zhuangzhuang Cui[1], Zihong Wang[1], Qi Dong[1], Jiajia Fan[1], Jinyu Jiang[1], Jun Ma[1], Zhihao Ma[1], Dazhuang Wang[1] & Xiaodi Ren [1] ✉

Uncontrollable zinc (Zn) plating and hydrogen evolution greatly undermine Zn anode reversibility. Previous electrolyte designs focus on suppressing $H_2O$ reactivity, however, the accumulation of alkaline byproducts during battery calendar aging and cycling still deteriorates the battery performance. Here, we present a direct strategy to tackle such problems using a strong Brønsted acid, bis(trifluoromethanesulfonyl)imide (HTFSI), as the electrolyte additive. This approach reformulates battery interfacial chemistry on both electrodes, suppresses continuous corrosion reactions and promotes uniform Zn deposition. The enrichment of hydrophobic $TFSI^-$ anions at the Zn|electrolyte interface creates an $H_2O$-deficient micro-environment, thus inhibiting Zn corrosion reactions and inducing a ZnS-rich interphase. This highly acidic electrolyte demonstrates high Zn plating/stripping Coulombic efficiency up to 99.7% at $1\,mA\,cm^{-2}$ ( > 99.8% under higher current density and areal capacity). Additionally, $Zn\,||\,ZnV_6O_9$ full cells exhibit a high capacity retention of 76.8% after 2000 cycles.

Grid-scale energy storage is essential for overcoming the intermittent and unstable nature of renewable energy generation[1]. Aqueous batteries have emerged as a promising technology for large-scale energy storage, offering unparalleled safety, reliability, and environmental friendliness[2,3]. Among the myriad of aqueous batteries, rechargeable aqueous zinc (Zn) batteries (RAZBs) garnered remarkable interest, propelled by the unique merits of Zn metal. This includes its high abundance, low cost, remarkable theoretical capacity ($820\,mAh\,g^{-1}$, $5855\,mAh\,cm^{-3}$), and low redox potential (−0.76 V vs. standard hydrogen electrode)[4–9].

In spite of the aforementioned advantages, the development of RAZBs faces significant challenges due to the reactive nature of the Zn anode in the aqueous electrolyte[10]. Its spontaneous reactions with $H_2O$ (self-corrosion) will cause hydrogen evolution reactions (HER) and the accompanying $OH^-$ formation would induce further adverse reactions, e.g., the precipitation of Zn hydroxide sulfate hydrate ($Zn_4SO_4(OH)_6 \cdot$

$xH_2O$), $Zn(OH)_2$, or ZnO on the surface[11–13]. This is closely related to the poor Coulombic efficiency (CE) and dendrite growth of Zn metal anode[14–18]. More importantly, the corrosion of Zn metal persists during the entire lifespan of RAZBs, not only during battery cycling but during idle times[19,20]. This issue is especially critical for grid-scale energy storage applications, as the batteries are expected to be operational for more than 20 years. Upon the contact between the Zn anode and electrolyte, self-corrosion automatically initiates, which generates $H_2$ and randomly distributes the above-mentioned alkaline byproducts during the rest period even before battery testing[11,21]. The presence of these byproducts masks the reaction sites on the Zn anode, leading to uneven Zn deposition during subsequent deposition processes and exacerbating other side reactions[11,22]. Furthermore, It is crucial to note that corrosion not only affects the Zn anode but also results in capacity degradation of cathode materials. The continuous release of $OH^-$ from Zn corrosion establishes a sustained concentration gradient of $OH^-$

[1]Hefei National Research Center for Physical Sciences at the Microscale, CAS Key Laboratory of Materials for Energy Conversion, Department of Materials Science and Engineering, University of Science and Technology of China, Hefei, Anhui 230026, China. ✉e-mail: xdren@ustc.edu.cn

ions, prompting their migration to the cathode side. This disrupts the pH balance of the electrolyte and causes the simultaneous accumulation of byproducts at the cathode surface, leading to capacity degradation in certain cathode materials[23]. As a result, the formation and migration process of OH[-] create a self-sustaining closed loop, thus continuously diminishing the battery performance.

To reduce the reactivity of $H_2O$ toward Zn metal, several electrolyte design strategies have been implemented[24-29]. Wang et al. proposed a high concentration aqueous electrolyte with 1 m $Zn(TFSI)_2$ + 20 m LiTFSI to minimize the amount of free $H_2O$ molecules in the electrolyte and realized a high Zn plating/ stripping CE of 99.7%[30]. With the use of a non-solvating diluent (1,4-dioxane), our group developed localized high-concentration aqueous electrolytes (LHCE) with improved rate capability and reduced cost[31,32]. Furthermore, highly effective fluorinated interphases (e.g., $ZnF_2$) were formed with enhanced anion interfacial chemistry in the LHCE solvation structure. Meanwhile, various organic co-solvents (e.g., dimethyl sulfoxide[15,33], ethylene glycol[34], ethylene glycol monomethyl ether[35], tetraglyme[36], etc.) have been incorporated into the $Zn^{2+}$ solvation structure to replace active $H_2O$ molecules. While greatly increased Zn plating/ stripping CEs have been achieved, there have also been concerns about battery safety with the use of flammable solvents. Recently, Wang et al. demonstrated an aqueous $ZnCl_2$ electrolyte by incorporating LiCl as a supporting salt, which enables highly efficient Zn plating/stripping[37]. In a separate study, Ji et al. introduced additional chloride salts (LiCl and trimethylammonium chloride (TMACl) and dimethyl carbonate into a concentrated $ZnCl_2$ electrolyte to significantly mitigate the HER side reactions[38].

While the prevalent studies focus on suppressing the reactivity of $H_2O$ to mitigate the spontaneous Zn corrosion reaction, there is a lack of effective strategies to tackle the deposited alkaline byproducts, which would worsen the instability of Zn metal during repeated cycling. To terminate the self-reinforcing cycle and ensure uniform Zn deposition, it is essential to eliminate these undesirable byproducts both during the resting period and throughout battery operation. Therefore, introducing acidic species into the electrolyte could be a direct and effective solution for the above problem. However, this approach has rarely been investigated due to concerns about severe hydrogen evolution side reactions.

Here, we propose to modulate the corrosion pathways of Zn metal by adding a strong Brønsted acid, bis(trifluoromethanesulfonyl)imide (HTFSI), into the conventional aqueous electrolyte (1 m $ZnSO_4$). The HTFSI additive could effectively prevent the accumulation of insoluble alkaline byproducts on the Zn surface, thus enabling subsequent uniform deposition of Zn. Furthermore, the hydrophobic TFSI[-] could accumulate at the Zn surface to mitigate $H_2O$ contact and promote the formation of ZnS-rich protective interphase, effectively inhibiting self-corrosion and HER. As a result, the electrolyte with the strong Brønsted acid demonstrates significantly improved reversibility of Zn anodes, with a high CE of ~ 99.7% and excellent stability of Zn plating/stripping in Zn||Cu cells. Moreover, Zn||Zn cells exhibit stable cycling over 2200 h at a high current density of 4 mA cm[-2] with an areal capacity of 4 mAh cm[-2]. The HTFSI additive also solves the alkaline corrosion issue to the cathode material and full cells with the zinc vanadate ($ZnV_6O_9$, ZVO) cathode demonstrate a capacity retention of 76.8% after 2000 cycles (~0.012% capacity decay per cycle). This work presents a perspective to modulate the Zn metal deposition behavior and interphase chemistry, unlocking the potential for highly efficient advanced RAZBs.

## Results and discussion
### Strategies for regulating self-corrosion of Zn anode
The reason for employing nonconventional Brønsted acidic additives for RAZBs lies in the unavoidable formation of alkaline byproducts on the Zn surface in typical neutral electrolytes. In conventional aqueous electrolytes (such as $ZnSO_4$), the self-corrosion reactions of Zn metal with or without $O_2$ all involve the participation of $H_2O$[10,27,29], which is the irreplaceable component for RAZBs.

$$\text{Without } O_2: Zn + 2H_2O \rightarrow Zn^{2+} + 2OH^- + H_2\uparrow \qquad (1)$$

$$\text{With } O_2: 2Zn + O_2 + 2H_2O \rightarrow 2Zn^{2+} + 4OH^- \qquad (2)$$

The enrichment of alkaline hydroxide anions at the Zn surface easily induces the deposition of poorly conductive byproducts, e.g., $Zn_4SO_4(OH)_6 \cdot xH_2O$ (ZSH) as in typical sulfate electrolytes (Fig. 1a). The byproducts generated during the resting period precipitate randomly

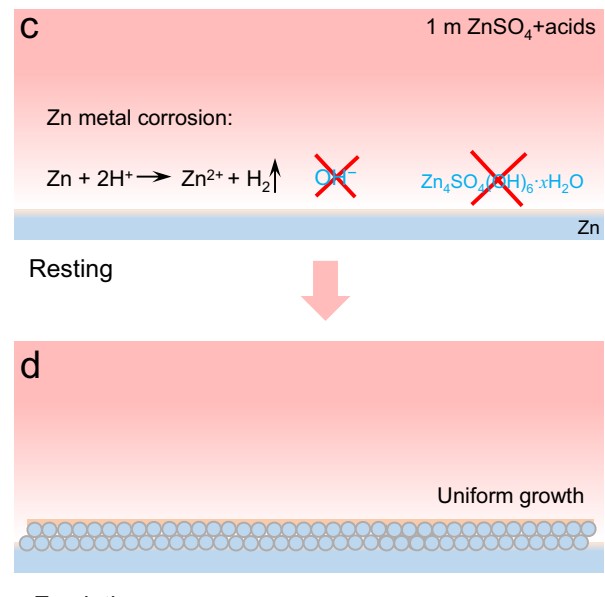

**Fig. 1 | Illustration of Zn surface reaction mechanism. a** Self-corrosion of Zn in 1 m $ZnSO_4$ during battery resting, and **b** subsequent Zn deposition process. **c** Self-corrosion of Zn in 1 m $ZnSO_4$ + acids during battery resting, and **d** subsequent Zn deposition process.

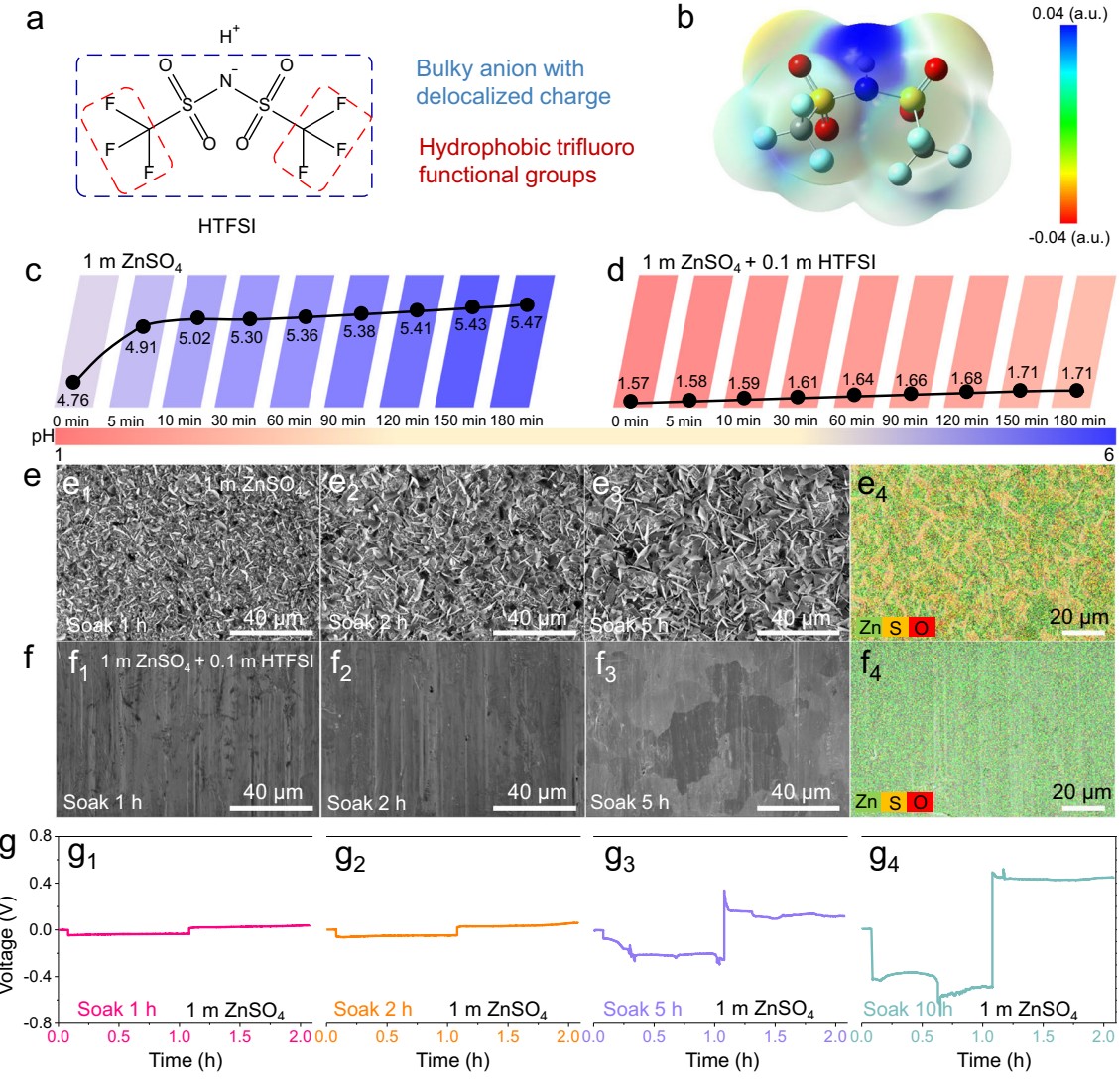

**Fig. 2 | Effects of self-corrosion on electrolyte pH, electrode surface products, and Zn deposition. a** The molecular structure of HTFSI. **b** The electrostatic potential map (ESP) of HTFSI. In situ pH monitoring of Zn||Zn cells during resting in different electrolytes, **c** 1 m ZnSO$_4$; **d** 1 m ZnSO$_4$ + 0.1 m HTFSI. **e** SEM images of Zn surfaces after soaking in the 1 m ZnSO$_4$ electrolyte for 1 h (**e$_1$**), 2 h (**e$_2$**), and 5 h (**e$_3$**); (**e$_4$**) EDS elemental mapping of Zn surfaces after soaking in the 1 m ZnSO$_4$ electrolytes for 5 h. **f** SEM images of Zn surfaces after soaking in the 1 m ZnSO$_4$ + 0.1 m HTFSI electrolyte for 1 h (**f$_1$**), 2 h (**f$_2$**), and 5 h (**f$_3$**); (**f$_4$**) EDS elemental mapping of Zn surfaces after soaking in the 1 m ZnSO$_4$ + 0.1 m HTFSI electrolyte for 5 h. **g** The voltage-time profiles of Zn||Zn cells, the Zn electrode was soaked in the 1 m ZnSO$_4$ electrolyte for 1 h (**g$_1$**), 2 h (**g$_2$**), 5 h (**g$_3$**), and 10 h (**g$_4$**) before use.

on the surface, leading to subsequent nonuniform Zn metal deposition and dendrite growth (Fig. 1b). This would greatly accelerate corrosion reactions, causing the accumulation of byproducts and further exacerbating the problem. To enable precise Zn metal electrodeposition, it becomes imperative to clean up the Zn metal surface. Brønsted acid, with its active proton-releasing ability, emerges as a promising candidate for this purpose. While acid pretreatments (e.g., HCl[39], H$_3$PO$_4$[40]) of Zn metal have been used to remove the oxide layer and create micro-structures on the Zn surface, the direct use of acidic additives in electrolytes is rare due to the concerns of severe hydrogen evolution side reactions. Chao et al. reported a significant improvement in the performance of an electrolytic Zn-MnO$_2$ battery by introducing H$_2$SO$_4$ into the electrolyte. However, their work only focused on the effect of H$_2$SO$_4$ on the cathode[41]. Recently, Li et al. introduced N,N-dimethylformamidium trifluoromethanesulfonate (DOTf) as an electrolyte additive, which forms both triflic acid (HOTf) and N,N-dimethylformamide (DMF) upon hydrolysis, and greatly improved the Zn metal CE. It raises an intriguing question about the Zn metal behavior with the presence of a neat and stronger Brønsted acid[42,43].

Although protons (H$^+$) could react with Zn metal (Fig. 1c), it would also avoid the formation of OH$^-$, thus inhibiting the precipitation of alkaline byproducts on the Zn surface and paving the way for the uniform deposition of Zn (Fig. 1d). In addition, it is possible to tune the reactivity and the concentration of the acidic species to mitigate their corrosion to Zn metal while taking advantages of their ability to clean-up the surface byproducts.

## Effect of self-corrosion during battery resting

HTFSI, which is one of the strongest Brønsted acid known-to-date[44,45], was chosen as the additive for this study to address the above-mentioned issues of Zn metal anode. As depicted in Fig. 2a, b, the TFSI$^-$ anion has a bulky structure, where the negative charge is highly delocalized. The weak cation-anion interactions greatly increase the Brønsted acidity of HTFSI. Moreover, the two per-fluorinated -CF$_3$ groups are oriented in opposite directions with respect to the central S−N−S unit, which endows the TFSI$^-$ anion with good hydrophobicity. Previous studies have shown that this hydrophobic property of TFSI$^-$ could be beneficial for the stability of metals against

moisture and aqueous solution[46,47]. We first explored its impact on Zn metal self-corrosion reactions during battery resting by in situ pH monitoring of the Zn||Zn cells. Detailed information on our test setup can be found in the supporting information (Supplementary Fig. 1 and 2)[48]. When immersing the Zn anode in the 1 m ZnSO₄ electrolyte for 180 min, we observed a significant increase in pH from 4.76 to 5.47. This pH shift to the alkaline direction would facilitate the formation of ZSH. The large fluctuations in pH indicate that the Zn anode underwent noticeable self-corrosion in the 1 m ZnSO₄ electrolyte (Fig. 2c). In contrast, we observed a slight increase in pH from 1.57 to 1.71 in the 1 m ZnSO₄ + 0.1 m HTFSI electrolyte, this indicates that the addition of HTFSI can maintain a relatively low pH at the Zn/electrolyte interphase (Fig. 2d). As shown in Supplementary Fig. 3a, we extended the rest time of the Zn/1 m ZnSO₄ + 0.1 m HTFSI/Zn cell and observed that the pH increased to 3.58 after a 10-hour rest. However, with longer cycling, the pH value is lower than that after resting for the same time (Supplementary Fig. 3b). The Zn/1 m ZnSO₄ + 0.1 m HTFSI/Zn cell maintains a relatively low pH during the cycling process, suggesting that the formation of a more uniform and protective SEI during the electrochemical process.

Figure 2e, f shows the typical scanning electron microscopy (SEM) images and energy dispersive X-ray spectrometry (EDS) mapping results of the Zn foils after self-corrosion tests for varied durations. When the Zn foil was immersed in the 1 m ZnSO₄ electrolyte for 1 h, a considerable amount of byproducts was observed on the Zn surface (Fig. 2e), which also further grew with the increase of contact duration. EDS mapping revealed that the Zn foil surface contained a significant amount of S and O species following immersion in the 1 m ZnSO₄ electrolyte for 5 h (Fig. 2e₄). X-ray diffraction (XRD) results confirmed that these species were mainly ZSH (Supplementary Fig. 4). By contrast, the addition of HTFSI results in a remarkably smooth Zn surface that avoids the production of ZSH byproducts (Fig. 2f₁-f₄; Supplementary Fig. 5). Furthermore, the influence of the ZSH formed during battery resting on subsequent electrochemical processes was investigated. The polarization of the Zn||Zn (Zn foils soaked in 1 m ZnSO₄ electrolyte before use) cell increased with prolonged soaking time, leading to abnormal voltage-time profiles and earlier battery failure (Fig. 2g₄, g₅ and Supplementary Fig. 6). Serious dendrite penetration into the separator was observed when the cells were disassembled (Supplementary Fig. 7). The experimental results confirm that self-corrosion byproducts adhered to the Zn electrode surface would severely influence Zn deposition. In contrast, HTFSI can modify the corrosion pathway of Zn anode during battery resting, prevent the formation of corrosion byproducts such as ZSH, and facilitate subsequent uniform Zn deposition (Supplementary Fig. 8). In addition, we also analyzed the corrosion behavior of Zn anodes in the electrolyte under O₂-saturated and O₂-free conditions. As shown in the SEM, EDS mapping, and XRD results (Supplementary Fig. 9), in the 1 m ZnSO₄ electrolyte, no matter whether O₂ exists or not, obvious Zn corrosion reactions occur, while O₂ greatly accelerates the corrosion process. Nevertheless, in the presence of HTFSI, even under O₂-saturated conditions, no accumulation of corrosion byproducts was observed on the Zn surface (Supplementary Fig. 9 and Fig. 10). Furthermore, we noticed ZSH formation even on copper surfaces when soaking a Cu foil in the 1 m ZnSO₄ electrolyte for 5 h. The EDS results in Supplementary Fig. 11 indicated that the hexagonal-shaped crystals are mostly likely ZSH. The formation of ZSH on the Cu surface is probably due to galvanic corrosion, and would have a negative influence on the plating/stripping process of Zn metal.

## Electrochemical performance of Zn anodes

The results of the electrochemical stability tests of the Zn anode are shown in Fig. 3. The electrolyte with 0.1 m HTFSI displays better Zn anode stability and reversibility among different concentrations of HTFSI (Supplementary Fig. 12). Adding too little HTFSI will have a limited effect while adding too much HTFSI will induce uncontrollable side reactions such as HER and Zn corrosion. The Zn plating/stripping behavior in 1 m ZnSO₄ + 0.1 m HTFSI was examined in Zn||Cu cells at 1 mA cm⁻² and 0.5 mAh cm⁻² (a frequently used test condition for Zn anode CE measurements)[42]. As depicted in Fig. 3a and Supplementary Fig. 13, the CE of the Zn electrode in 1 m ZnSO₄ + 0.1 m HTFSI electrolytes exhibits rapid stabilization, reaching 99% within the initial 40 cycles. Subsequently, it achieved a CE of 99.7% over 1400 cycles. In contrast, the baseline electrolyte of 1 m ZnSO₄ (Fig. 3a) shows irregularly fluctuating CEs and quickly fails after 180 cycles. In addition, the initial CEs in 1 m ZnSO₄ and 1 m ZnSO₄ + 0.1 m HTFSI are 86.48% and 93.27%, respectively (Supplementary Fig. 14). The addition of HTFSI greatly improves the initial CE of Zn plating/stripping. This indicates that the removal of the alkaline corrosion byproducts during the rest period is highly beneficial for improving the Zn metal reversibility by controlled Zn deposition. Moreover, under 4 mA cm⁻² current density and 2 mAh cm⁻² areal capacity, the designed electrolyte demonstrates an impressive CE exceeding 99% within just 10 cycles, which is sustained for 550 cycles (up to ~99.85%) (Fig. 3b). Figure 3c illustrates that the nucleation over-potential ($\eta_n$) and growth over-potential ($\eta_g$) on the Cu substrate decreases with the introduction of HTFSI, indicating a reduced barrier for Zn deposition. From the cyclic voltammetry (CV) curves in Supplementary Fig. 15, it is also clear that the addition of HTFSI reduces nucleation overpotential. The kinetics of Zn deposition were quantitatively evaluated from the activation energy ($E_a$) through the Arrhenius equation. Among the tested electrolytes, 1 m ZnSO₄ + 0.1 m HTFSI electrolyte shows a lower $E_a$ value of 47.88 kJ mol⁻¹(Supplementary Fig. 16). It was confirmed again that the addition of HTFSI can improve the kinetics of the electrode reaction. Additionally, chronoamperometry was used to assess the effect of the HTFSI additive on the nucleation and growth of Zn metal (Supplementary Fig. 17). The presence of HTFSI leads to a more stable current, indicating of 3D diffusion for uniform crystal growth. Supplementary Fig. 18 displays the addition of HTFSI in aqueous ZnSO₄ electrolyte results in an increase of corrosion potential from −0.99 to −0.97 V vs. Ag/AgCl and a significant decrease of corrosion current density from 1.62 to 0.64 mA cm⁻². The more positive corrosion potential and lower corrosion current density indicate less tendency of corrosion reaction and low corrosion rate, respectively. The electrochemical tests of Zn||Zn and Zn||Ti cells in different electrolytes also support the improved compatibility of Zn anode with the addition of HTFSI (Supplementary Fig. 19, 20). In addition, we also compared the differences between traditional acid pre-treatment methods and acid additive strategies. As shown in Supplementary Fig. 21, compared with the traditional acid pre-treatment method of Zn foil, the addition of HTFSI in the electrolyte greatly improves the CE of Zn plating/stripping during long-term cycling.

In addition, The Zn||Zn cell with HTFSI exhibits a stable polarization of ~30.5 mV and ultra-long cycling life of 2200 h, nearly 20 times longer than the cell without HTFSI at a current of 4 mA cm⁻² and a capacity of 4 mAh cm⁻² (Fig. 3e). It is likely that the removal of ZSH provides more reaction sites for the deposition of Zn, which is favorable for the deposition of Zn. The rate capability is evaluated by gradually increasing the current density (0.5 mA cm⁻²) with a fixed plating/stripping time (0.5 h). As shown in Fig. 3d, the HTFSI additive significantly improved the rate performance, exhibiting stable discharge and charge voltage profiles at current densities up to 17 mA cm⁻². A critical parameter of Zn anodes for estimating the Zn anode energy density and cycling performance is the depth of discharge (DOD$_{Zn}$), referring to the fraction of Zn in a Zn anode that takes part in redox reactions during charging/discharging. As shown in Supplementary Fig. 22, the Zn||Zn (50 μm thickness) cell with the HTFSI additive could keep a stable cycle under 10 mA cm⁻²/10 mAh cm⁻² over 100 h, corresponding to a DOD$_{Zn}$ of over 34%. Moreover, the Zn||Zn (10 μm thickness) cell at 2 mA cm⁻²/2 mAh cm⁻² keeps stable cycling over

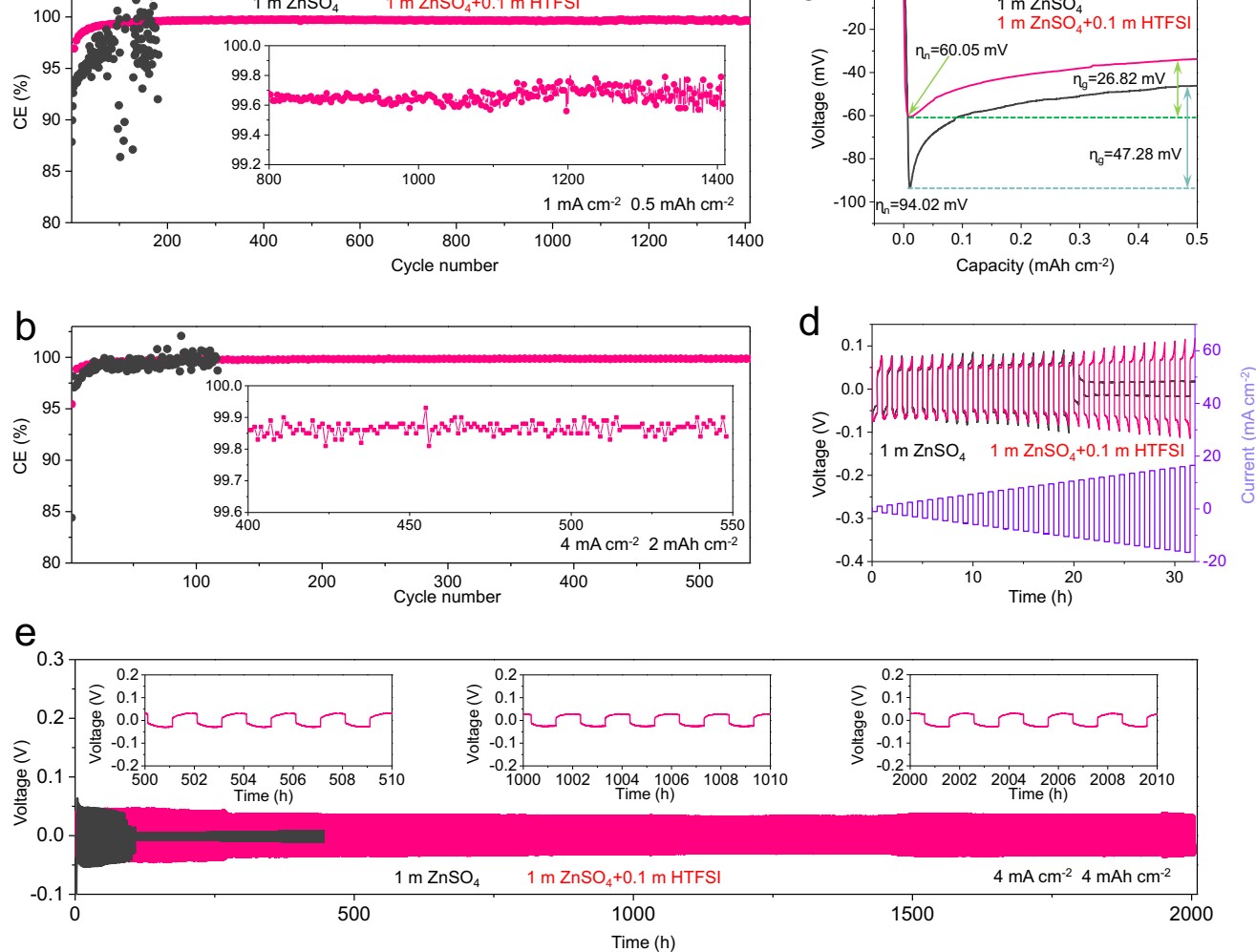

**Fig. 3 | Electrochemical tests studies of Zn plating/stripping in different electrolytes. a** and **b** Zn CE evolution in Zn||Cu cells with different electrolytes at 1 mA cm$^{-2}$, 0.5 mAh cm$^{-2}$ (**a**) and 4 mA cm$^{-2}$, 2 mAh cm$^{-2}$ (**b**), respectively. **c** Nucleation and growth over-potential on the Cu substrate. **d** Voltage evolution of Zn||Zn cells at step-increased current densities. **e** Galvanostatic Zn plating/stripping in Zn|1 m ZnSO$_4$ + 0.1 m HTFSI|Zn and Zn|1 m ZnSO$_4$ | Zn cells at 4 mA cm$^{-2}$ and 4 mAh cm$^{-2}$.

1000 h (Supplementary Fig. 23), corresponding to a DOD of over 34%. The impedance spectra of the Zn||Zn cell before and after cycling are shown in Supplementary Fig. 24, and the impedance of the Zn|1 m ZnSO$_4$ + 0.1 m HTFSI|Zn system is smaller than that of the Zn|1 m ZnSO$_4$ | Zn system. Therefore, the introduction of HTFSI into the electrolyte showed distinct advantages for Zn anode in terms of rate capability, deposition barrier, and cycling stability. Here, an important consideration is whether the introduction of HTFSI will cause significant hydrogen evolution. To verify this, we used gas chromatography to detect hydrogen evolution during cycling (Supplementary Fig. 25). The results showed that the introduction of HTFSI actually reduced the amount of hydrogen evolution. It is possible that the introduction of HTFSI promoted the uniform deposition of Zn and modulated the interphase properties to suppress the side reactions. Furthermore, our tests using ZnCl$_2$ and Zn(CH$_3$COO)$_2$ as the Zn salts also demonstrated improved Zn CEs in Zn||Cu cells with the addition of 0.1 m HTFSI, as shown in Supplementary Fig. 26. This indicates the general positive effect of HTFSI on the Zn anode reversibility. Their differences of Zn CEs compared to that in the ZnSO$_4$-based electrolyte could be mainly attributed to the SEI composition, which will be discussed in the later session.

## Characterizations of the electrodes

To understand the impact of the HTFSI additive, we characterized the morphology of deposited Zn metal on Cu substrate (after 10 cycles with a current density of 1 mA cm$^{-2}$ and an areal capacity of 0.5 mAh cm$^{-2}$). For easier cross-section observation, the deposition capacity was adjusted to 2 mAh cm$^{-2}$, and the current density was set at 4 mA cm$^{-2}$. In the 1 m ZnSO$_4$ + 0.1 m HTFSI electrolyte, we found dense, uniform, nanosized Zn deposits (Fig. 4a) with a thickness of 4.3 μm, compared to the moss-like Zn growth with a thickness of 14.8 μm in the 1 m ZnSO$_4$ electrolyte (Fig. 4a$_4$, b$_4$). Furthermore, EDS mapping images (Fig. 4a$_{a5}$, a$_6$, and Supplementary Fig. 27) demonstrated a homogeneous distribution of a large amount of Zn (73.89%), along with small amounts of S (0.80%) and O (25.31%) elements on the Zn surface in the 1 m ZnSO$_4$ + 0.1 m HTFSI system. The very limited S content indicates that ZSH is not accumulated on the Zn surface. XRD results confirmed that ZSH was not detected in the 1 m ZnSO$_4$ + 0.1 m HTFSI system (Supplementary Fig. 28). In contrast, after Zn was cycled in the 1 m ZnSO$_4$ electrolyte, a substantial amount of S (4.71%) and O (61.58%) elements showed uneven distribution on the Zn surface (Fig. 4b$_5$, b$_6$, and Supplementary Fig. 29), with S and O appearing in the same region, indicating the potential formation of ZSH in this area. XRD

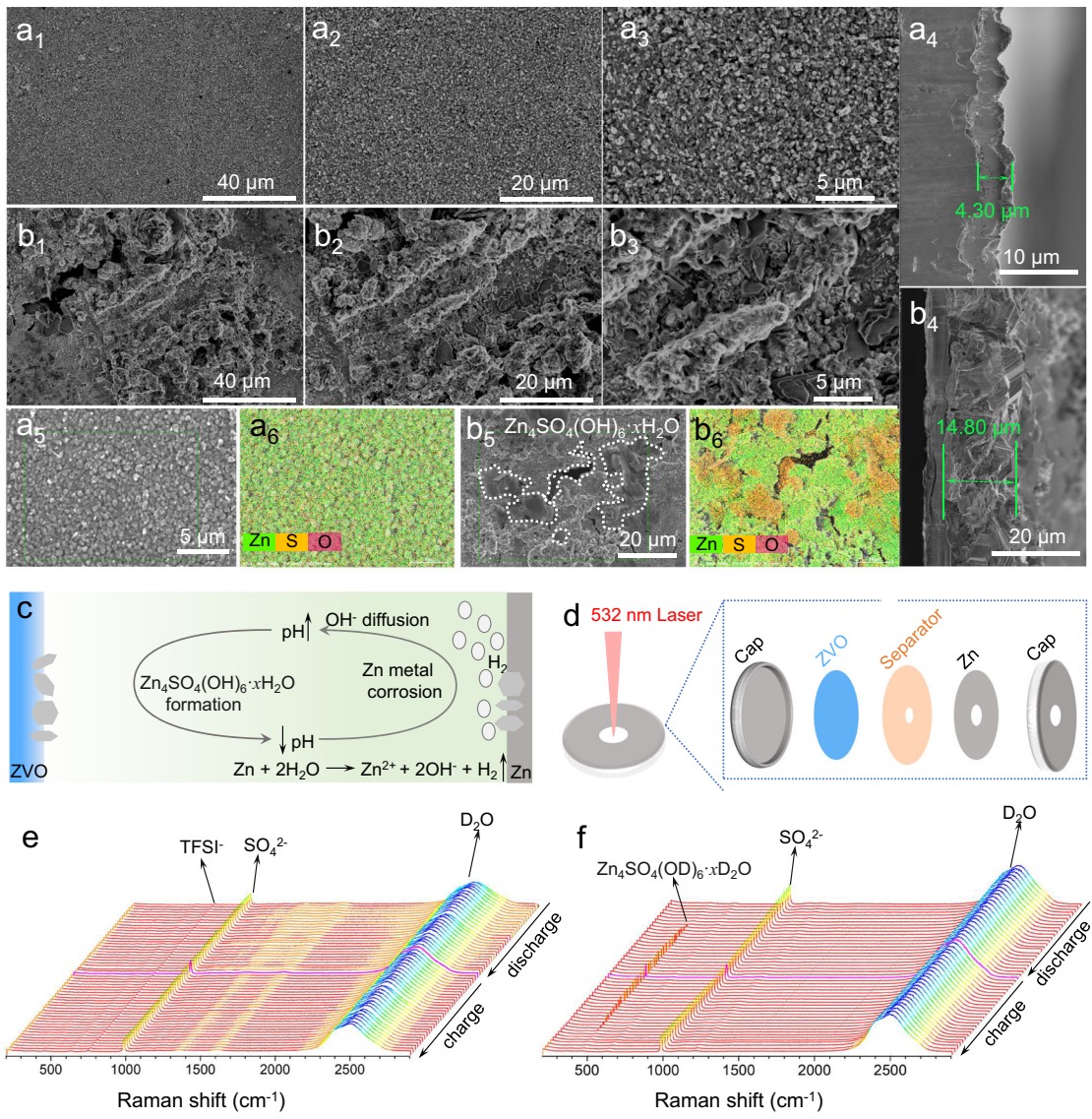

**Fig. 4 | Characterizations of the anode and the cathode.** Morphologies of plated Zn on Cu. SEM images of plated Zn in 1 m ZnSO$_4$ + 0.1 m HTFSI, ($a_1$-$a_3$ and $a_5$) top view and ($a_4$) cross-sectional view; SEM images of plated Zn in 1 m ZnSO$_4$, ($b_1$-$b_3$ and $b_5$) top view and ($b_4$) cross-sectional view. Inside the white dotted line is Zn$_4$SO$_4$(OH)$_6$·$x$H$_2$O. EDS elemental mapping of Cu electrodes surface after plated Zn on Cu in 1 m ZnSO$_4$ + 0.1 m HTFSI ($a_6$) and 1 m ZnSO$_4$ ($b_6$) electrolytes, respectively. **c** Formation mechanism of byproducts on the cathode surface. **d** Schematic diagram of the in situ Raman test setup. e and f) In situ Raman spectrum of ZVO in 1 m ZnSO$_4$ + 0.1 m HTFSI (**e**) and 1 m ZnSO$_4$ (**f**).

results further confirmed the presence of ZSH in the 1 m ZnSO$_4$ electrolyte (Supplementary Fig. 30).

As Choi's group has recently confirmed that alkali corrosion is the origin of the decay of vanadium oxide-based RAZBs[23], the effect of the Zn corrosion on the cathode cannot be ignored. As shown in Fig. 4c, the OH$^-$ formed by Zn corrosion diffuses to the cathode, which will form ZSH on the cathode surface and damage the cycle stability of the battery. In situ Raman is an excellent technique to study changes in the cathode surface in real time due to the short spectrum acquisition time (~60 s per spectrum). The schematic diagram of the in situ Raman device is shown in Fig. 4d. In the presence of HTFSI, in situ Raman results showed no obvious change on the cathode surface (Fig. 4e), while the appearance and disappearance of ZSH were observed in the 1 m ZnSO$_4$ system (Fig. 4f). This suggests the formation of ZSH is due to the diffusion of OH$^-$ into the vicinity of the cathode material. With the addition of HTFSI, the protons effectively remove excessive OH$^-$, thus preventing the participation of ZSH. To sum up the above results, the formation of corrosion byproduct ZSH was not detected on either anode or cathode after the introduction of HTFSI.

## Electrochemical performance of Zn||ZVO full cells

The effect of 1 m ZnSO$_4$ + 0.1 m HTFSI on the full cell electrochemical performance was studied in a coin cell configuration at room temperature, using a ZnV$_6$O$_9$ (ZVO) cathode (the ZVO relevant information can be found in Supplementary Fig. 31). First, the electrochemical performance of ZVO cathode in different electrolytes was investigated using CV at 1 mV s$^{-1}$ in Zn||ZVO coin cells. ZVO in both electrolytes featured two redox peaks (Fig. 5a). Figure 5b presents typical charge and discharge profiles of the ZVO for the first 1000 cycles at a current density of 2 A g$^{-1}$ in the 1 m + 0.1 m HTFSI electrolyte. The ZVO nanobelts deliver an average operating voltage of about 0.8 V vs. Zn$^{2+}$/Zn, as well as a high reversible capacity of 237 mAh g$^{-1}$ (based on the mass of ZVO) in the first cycle. In the 1 m ZnSO$_4$ electrolyte, a dramatic capacity loss was observed with cycling (Fig. 5c), which is likely due to the formation of ZSH on both electrodes. As illustrated in Fig. 5d, Zn|

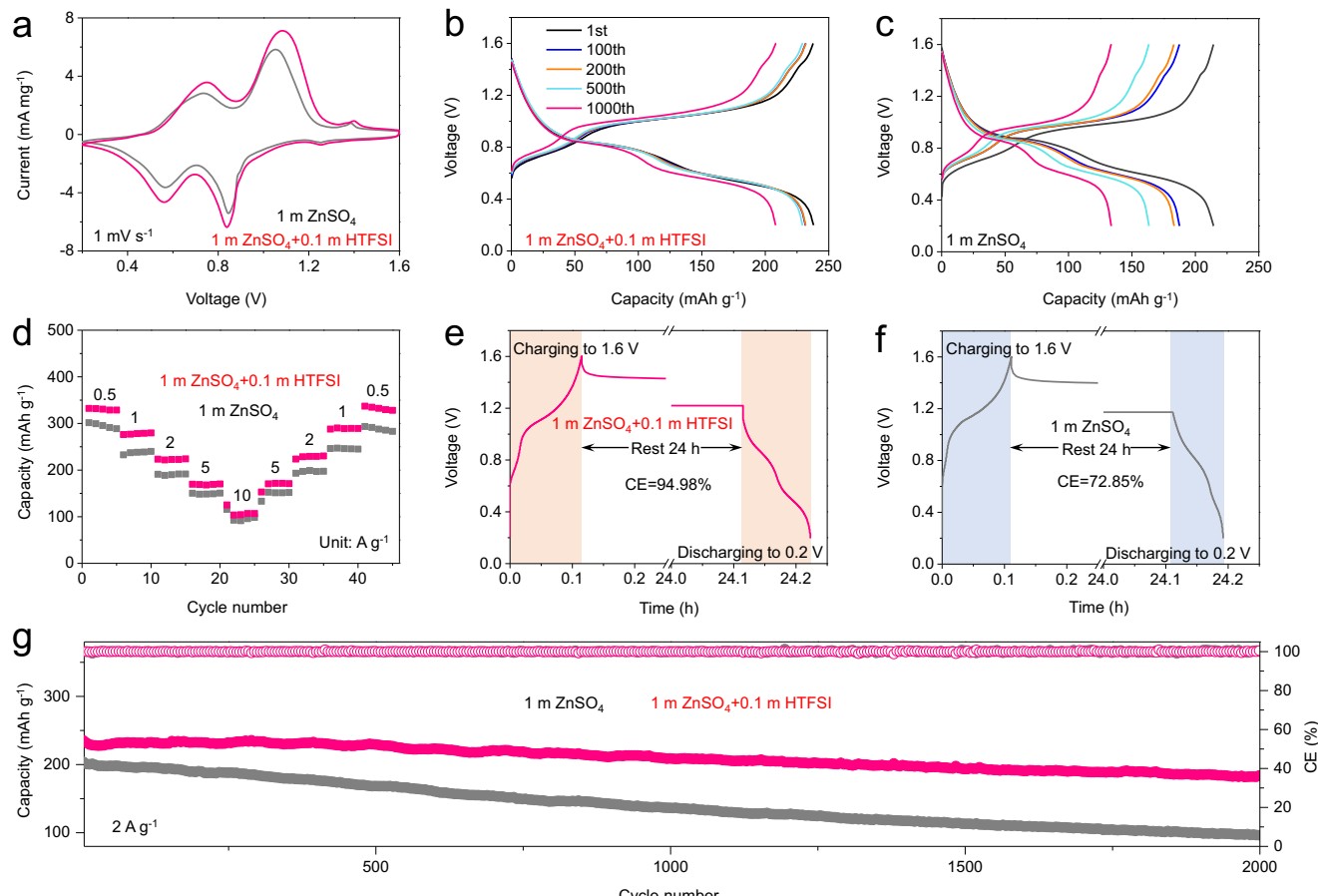

**Fig. 5 | The electrochemical performance of Zn||ZVO full cells. a** The CV profiles of full cells in different electrolytes at 1 mV s⁻¹. **b** The voltage profiles of full cells in 1 m ZnSO₄ + 0.1 m HTFSI electrolyte. **c** The voltage profiles of full cells in 1 m ZnSO₄ electrolyte; **d** Rate performance. **e** and **f** Self-discharge tests of the full cells after 24 h rest. **g** Long-term cycling stability of full cells in different electrolytes at 2 A g⁻¹.

1 m + 0.1 m HTFSI | ZVO displays higher specific capacities than Zn|1 m ZnSO₄ | ZVO at different current densities. Even at a high current density of 10 A g⁻¹, a high specific capacity of 108.5 mAh g⁻¹ was still obtained for Zn|1 m ZnSO₄ + 0.1 m HTFSI | ZVO. In addition, the self-discharge behaviors of the full cells were explored. After resting for 24 h, Zn|1 m ZnSO₄ + 0.1 m HTFSI | ZVO holds ≈94.98% of its original capacity, which is much higher than that of Zn|1 m ZnSO₄ | ZVO (72.85%) (Fig. 5e, f). Besides the excellent rate performance, the cycling performance of Zn|1 m ZnSO₄ + 0.1 m HTFSI | ZVO was also greatly improved. The cycling performance and corresponding CE of the cells are plotted in Fig. 5g. After 2000 cycles at 2 A g⁻¹, the cell with HTFSI retained a capacity of 182 mAh g⁻¹, much higher than the cell without the additive. Its capacity retention reached 76.8% and a stable CE of close to 100% was achieved.

## The working mechanism of HTFSI

As indicated by the Raman spectrum shown in Supplementary Fig. 32, the S-O vibration signals related to SO₄²⁻ and the O-H vibration signal associated with H₂O exhibit no apparent changes after the addition of HTFSI. This suggests that the effect of HTFSI cannot be attributed to the change in the electrolyte solvation structure. To delve into the functioning mechanism of the HTFSI additive, X-ray photoelectron spectroscopy (XPS) analysis of Zn electrodes was carried out at different depths of 0, ~2, ~10, and ~20 nm by Ar⁺ sputtering. All binding energies were calibrated using the C 1s peak (284.8 eV) as the reference. XPS spectra of the soaked Zn electrode in 1 m ZnSO₄ + 0.1 m HTFSI are shown in Fig. 6a. The S 2*p* spectrum confirms the existence of ZnSO₄ (169.4 eV) along with ZnSO₃ (167.7 eV) and ZnS species

(162.1 eV) on Zn electrode surfaces. This result indicates that Zn metal will spontaneously form ZnS in 1 m ZnSO₄ + 0.1 m HTFSI system. The ZnS component probably originates from the reduction of SO₄²⁻ rather than the TFSI⁻ anion owing to the lack of fluorine-related species in the F 1s XPS spectrums (Supplementary Fig. 33). Previous reports also showed that the reduction of SO₄²⁻ to ZnS could be induced in the presence of another superacid, triflic acid (HOTf), further confirming that HTFSI as a superacid is the key to promoting the formation of ZnS[42,43]. This conclusion was also supported by experimental evidence (Supplementary Fig. 34). In our designed experiment, we observed insoluble ZnS solid formation on Zn surface when in contact with HTFSI and 1 m ZnSO₄ simultaneously, as revealed by the XRD test (Supplementary Fig. 34b). However, the direct mixing of HTFSI and 1 m ZnSO₄ did not produce any insoluble solids (Supplementary Fig. 34c), indicating Zn is necessary for the formation of ZnS. In addition, XPS tests of the cycled Zn electrode in 1 m ZnSO₄ + 0.1 m HTFSI electrolyte also showed the presence of ZnSO₄, ZnSO₃, and ZnS at the surface (Fig. 6b). The sulfate signal was attributed to the precipitation of Zn salt in the electrolyte. The intensity of the ZnSO₄ signal decreased upon further sputtering, while the ZnS and ZnSO₃ signals increased, indicating the enrichment of ZnS in the inner SEI of the Zn anode. We also employed a non-destructive characterization method, grazing incidence X-ray diffraction (GIXRD), to identify the existence of ZnS. With the incident X-ray nearly parallel to the sample surface, GIXRD is particularly suitable for characterizing the surface structure. As illustrated in Supplementary Fig. 35, the ZnS signal could be detected through GIXRD. This additional evidence further supports that ZnS is actually formed on the Zn surface, not due to Ar⁺ sputtering. However,

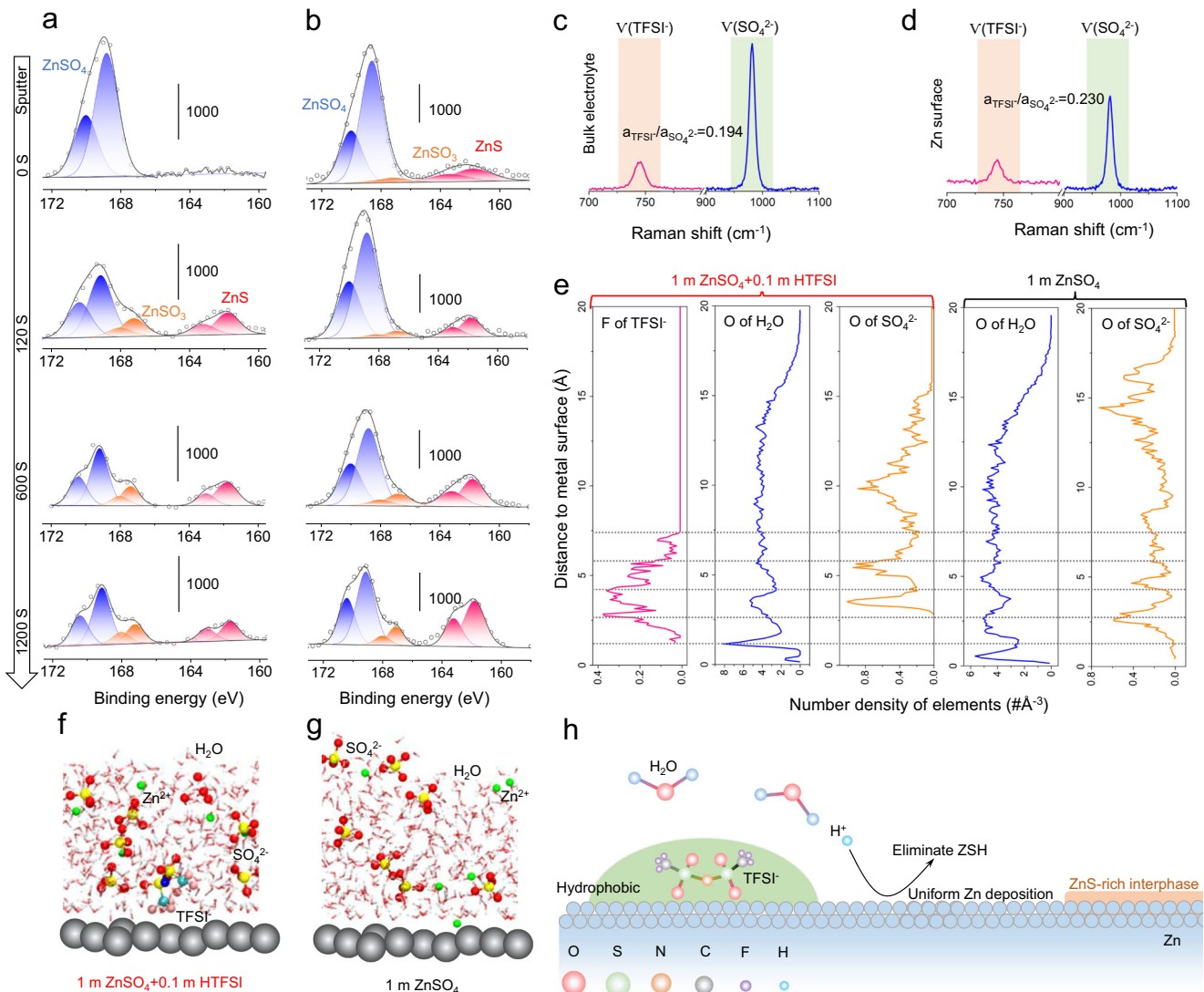

**Fig. 6 | Interfacial studies of the Zn|electrolyte interface in the 1 m ZnSO₄ + 0.1 m HTFSI electrolyte.** **a** XPS of Zn anode after soaking in 1 m ZnSO₄ + 0.1 m HTFSI for 5 h. **b** XPS of Zn anodes after 50 cycles in 1 m ZnSO₄ + 0.1 m HTFSI. **c** Raman spectrum of the bulk 1 m ZnSO₄ + 0.1 m HTFSI electrolyte and **d** Raman spectrum of the Zn surface region. $a_{TFSI^-}$ and $a_{SO_4^{2-}}$ are the peak areas of the TFSI⁻ and SO₄²⁻ signals, respectively. **e** Distribution of elements at different depths from the Zn surface and the formation of an H₂O-deficient region. Snapshots of the distribution of electrolyte species in **f** 1 m ZnSO₄ + 0.1 m HTFSI and **g** 1 m ZnSO₄ electrolytes. **h** Schematic illustration of the working mechanism of the HTFSI additive.

no discernible ZnS was detected from GIXRD on the Zn surfaces immersed in 1 m ZnCl₂ + 0.1 m HTFSI and 1 m Zn(CH₃COO)₂ + 0.1 m HTFSI electrolytes (Supplementary Fig. 36). This further confirms that the formation of ZnS from SO₄²⁻. While ZnF₂ was typically regarded as the favorable SEI component for the Zn anode, as an analogy to the Li anode, for inhibiting electrolyte side reactions and dendrite growth, it is noteworthy that ZnF₂ has a much higher solubility in water than that of ZnS (solubility constant $K_{sp} = 3.04*10^{-2}$ vs $2.5*10^{-22}$, 25 °C)[49,50]. Therefore, a ZnS-enriched SEI layer could potentially be more effective in isolating the active Zn from electrolyte corrosion.

Furthermore, the unique effect of HTFSI was revealed by comparison with the conventional sulfuric acid (H₂SO₄), hydrochloric acid (HCl), phosphoric acid (H₃PO₄) and the HOTf. The absence of ZSH on Zn metal immersed in the 1 m ZnSO₄ + 0.05 m H₂SO₄ electrolyte indicates that the introduction of protons into the electrolyte generally prevents the formation of alkaline byproducts (Supplementary Fig. 37 and 38). This observation is similar to that in the HTFSI system, indicating that protons can effectively scavenge alkaline byproducts on the Zn surface. However, we found that ZSH

could still be enriched on the Cu substrate after cycling in Zn||Cu cells with the 1 m ZnSO₄ + 0.05 m H₂SO₄ electrolyte (Supplementary Fig. 39). In addition, the Zn plating/stripping CE(<99.4%) and the cycle life (<200 cycles) are far inferior to those in the 1 m ZnSO₄ + 0.1 m HTFSI (Supplementary Fig. 40). The addition of HOTf additive (1 m ZnSO₄ + 0.1 m HOTf) also increases the Zn CE to ~99.6%, but the improvement in cycle life is still limited (Supplementary Fig. 41). As illustrated in Supplementary Fig. 42, the test results with other acids (HCl and H₃PO₄) indicate that a suitable amount of acids can enhance the Zn CE in Zn||Cu cells. Their differences with HTFSI are likely due to the properties of anions. However, it should also be noted that TFSI⁻ anion alone cannot afford reversible Zn metal cycling. The inferior Zn anode performances with only H⁺ or TFSI⁻ additive (Supplementary Fig. 40 and 43) suggest the synergistic effect between H⁺ and TFSI⁻.

Surface-enhanced Raman spectroscopy (SERS) and theoretical calculations were employed to gain a better understanding of the role of HTFSI in Zn metal stability. We compared the Raman signal of the bulk 1 m ZnSO₄ + 0.1 m HTFSI electrolyte with the signal at the Zn

surface region. As shown in Fig. 6c, d, compared with the bulk electrolyte, the peak area ratio of TFSI$^-$/SO$_4^{2-}$ at the Zn surface is higher (0.230 vs. 0.194), indicating that TFSI$^-$ anions prefer to be enriched on the electrode surface. Ab initio molecular dynamics (AIMD) simulations were also performed with explicit H$_2$O solvent molecules to understand the distribution of species at the interface. The number density distribution of H$_2$O, SO$_4^{2-}$, and TFSI$^-$ at different distances from the Zn surface are shown in Fig. 6e. The snapshots from the MD simulations are shown in Fig. 6f, g. In contrast to SO$_4^{2-}$ anion, TFSI$^-$ anion was found to be enriched on the Zn surface in the 1 m ZnSO$_4$ + 0.1 m HTFSI. The addition of HTFSI into the 1 m ZnSO$_4$ electrolyte also results in a decrease in the distribution of H$_2$O at the Zn surface due to the hydrophobic characteristics of TFSI$^-$. These results agree with the previous study that SO$_4^{2-}$ is hydrophilic, while TFSI$^-$ has pronounced hydrophobicity[46]. As a result of the strong hydrophobicity of TFSI$^-$, a H$_2$O-deficient region was formed on the Zn surface with greatly suppressed corrosion reactions. Therefore, the mechanism of action of HTFSI during the resting period of the battery can be attributed to two aspects. Firstly, the addition of strong Brønsted acid prevents the accumulation of corrosion products on the Zn surface. Secondly, the presence of HTFSI on the Zn surface creates a hydrophobic interfacial region with superacidic activity, inducing the formation of ZnS and preventing further corrosion of the Zn electrode (Fig. 6 h). These two aspects are both closely related to the property of the TFSI$^-$ anion, as the bulky structure with hydrophobic -CF$_3$ moieties not only delocalizes the negative charge to increase the proton donating ability but fosters a hydrophobic interface at the Zn surface to suppress further electrolyte corrosions.

In summary, this study challenges the conventional belief that acids would degrade the performance of Zn anodes. Instead, using the strong Brønsted acid (HTFSI) with hydrophobic moieties as the electrolyte additive for RAZBs effectively promotes the uniform deposition of Zn after removing insoluble alkaline byproducts at the Zn surface. The results are highly encouraging, as the preferred 1 m ZnSO$_4$ + 0.1 m HTFSI electrolyte demonstrated greatly improved performance: stable Zn plating/stripping for over 1400 cycles with a CE reaching 99.7% (>99.8% under higher rate and areal capacity), prolonged cycling stability of Zn||Zn cells for more than 1000 h under deep-discharge conditions (34% DOD$_{Zn}$), and stable Zn||ZVO full cell performance. The excellent Zn stability and reversibility are attributed to the direct function of strong Brønsted acid on the inhibition of alkaline ZSH formation, and the effect of hydrophobic TFSI$^-$ anion for inducing a ZnS-rich interfacial protective layer on the Zn surface, which effectively preventing continuous side reactions. This work unveils an intriguing discovery that strong and hydrophobic acids can yield beneficial effects on Zn anodes and provides valuable insights into the development of reversible Zn metal anodes.

## Methods
### Materials
Zinc sulfate heptahydrate (ZnSO$_4$·7H$_2$O), polytetrafluoroethylene preparation (PTFE, (C$_2$F$_4$)$_n$, 60 wt%) were purchased from Shanghai Aladdin Biochemical Technology Co., Ltd. Bis(trifluoromethanesulfonyl)imide (HTFSI, C$_4$F$_6$N$_1$O$_4$S$_2$H), vanadium pentoxide (V$_2$O$_5$) were purchased from Shanghai Macklin Biochemical Co., Ltd. Zinc acetate dihydrate (Zn(CH$_3$COO)$_2$·2H$_2$O), hydrogen peroxide (H$_2$O$_2$, 30 wt%), sulfuric acid (H$_2$SO$_4$, 98 wt%) were purchased from Sinopharm Chemical Reagent Co., Ltd. carbon black (Super C65) was purchased from Shanghai Haily Scientific & Trading Co., Ltd. All chemicals were all analytical grade and without further purification. Glass microfiber filters (Whatman GF/D) was purchased from Guangdong Canrd New Energy Technology Co., Ltd. Zn (10 μm, 50 μm) foil, Cu foil, and Ti mesh were obtained from Shenzhen Kejing Zhida Technology Co., Ltd.

### Preparation of electrolytes
Deionized (DI) water obtained from an ultrapure water production system (HHitech) was used to prepare all aqueous electrolytes. ZnSO$_4$·7H$_2$O was dissolved in deionized water to prepare 1 m ZnSO$_4$ electrolyte. Stoichiometric amount of HTFSI (0.01 m, 0.02 m, 0.05 m, 0.1 m, 0.2 m, and 0.5 m) was dissolved into the 1 m ZnSO$_4$ electrolytes to prepare HTFSI-containing electrolytes.

### Synthesis of ZVO
0.50 g of V$_2$O$_5$ was first added to a mixture of deionized water (7.7 g) and hydrogen peroxide (2.5 g). After stirring for 15 min, 40 g of deionized water was added, and the solution was thoroughly mixed again. The homogeneous dark red solution was then sonicated for 2 h. After 0.15 g of Zn(CH$_3$COO)$_2$·2H$_2$O was added to the above solution, and the solution was thoroughly remixed. The mixture was transferred to a sealed Teflon vessel and kept at 180 °C for 90 min. The final ZnVO nanostaves were obtained by centrifugal, washing with deionized water and ethanol, and drying in a 60 °C oven for 24 h.

### Electrochemical measurements
Cyclic Voltammetry (CV). CV curves tests were carried out using a BioLogic electrochemical working station (VMP3) at different scan rates and voltage windows. ZVO electrodes were used as the working electrode, and Zn foil was employed as both the reference and counter electrodes. All the measurements were carried out at room temperature; Electrochemical Impedance Spectroscopy (EIS) was measured on a BioLogic electrochemical working station (VMP3) in a frequency range from 0.1 Hz to 1 MHz with a sinus amplitude V$_a$ = 10 mV. Two-electrode setup of Zn||Zn cells was used to perform the test. The measurements were carried out at different temperatures (0 °C, 5 °C, 10 °C, 15 °C, 20 °C, 30 °C); Chronoamperograms (CA). CA tests were obtained on a BioLogic electrochemical station (VMP3). Two-electrode setup of Zn||Zn cells was used to perform the test. All the measurements were carried out at room temperature; Tafel Plot curves were conducted in a standard three-electrode configuration, in which Zn foil was used as the working and counter electrodes, and an Ag/AgCl electrode (saturated potassium chloride aqueous solution) was used as the reference electrode. TP curves tests were obtained on a BioLogic electrochemical working station (VMP3) in a voltage range from −0.25 V to 0.25 V vs. E$_{oc}$ at scan E$_{we}$ with d$_E$/d$_t$ = 1 mV s$^{-1}$.

### Ion conductivity measurements
The ionic conductivity measurements were conducted in a micro-electrochemical cell (electrolyte volume -1 mL, Supplementary Fig. 44), with a two-electrode setup (titanium foils as the electrodes). The impedance of the target electrolyte was first measured using EIS (100 mHz-1 MHz, 10 mV). Then the standard 1 M KCl electrolyte was tested as a reference (0.1118 S cm$^{-1}$, 25 °C) to calculate the ionic conductivity of the target electrolytes. The resulting ionic conductivity data is depicted in Supplementary Fig. 45. Notably, the ionic conductivity of the 1 m ZnSO$_4$ + 0.1 m HTFSI electrolyte surpasses that of the 1 m ZnSO$_4$ electrolyte. This enhancement can be attributed to the increased concentration of highly conductive protons in the electrolyte due to the addition of HTFSI.

### Zn||Zn cells and Zn||Cu cells assembly
The Zn||Zn cells and Zn||Cu cells were assembled with 2032-type coin cells for evaluating the electrochemistry process of Zn plating/stripping. Electrochemical stability was tested using Zn||Zn cells at room temperature. The Coulombic efficiency (CE) was tested using Zn||Cu cells to measure the reversibility of Zn plating/stripping process at different current densities with a cut-off voltage of 0.5 V. The Zn||Zn and Zn||Cu cells were assembled using Zn (Φ12 mm) and Zn (Φ12 mm) ||Cu (Φ16 mm) electrodes, The glass microfiber filters as the separator, and the 1 m ZnSO$_4$ and 1 m ZnSO$_4$ + x m HTFSI (x = 0.01, 0.05, 0.1, 0.2,

0.5, 1.0) as electrolyte. The electrolyte amounts are around controlled at 100 μL. These cells were tested on a Neware battery cycler (CT-4008T-5V10Ma-164) at room temperature. Note: This paper uses two thicknesses (10 μm, 50 μm) of Zn foil and 10 μm thick Zn foil only for the DOD test.

## Zn||ZVO batteries assembly

The Zn/ZVO batteries were composed of ZVO cathodes, glass fiber separators, Zn anodes, and the electrolyte (1 m $ZnSO_4$ and 1 m $ZnSO_4$ + 0.1 m HTFSI, 100 μL). the cathode was obtained by pressing the mixture of ZVO (70 wt%), Super C65 (20 wt%), and PTFE (10 wt%) on Ti mesh (Φ12 mm), and dried in a vacuum oven at 80 °C for 12 h. The areal mass loading of ZVO was around 2 mg cm$^{-2}$. The specific capacity of Zn (50 μm)/ZVO batteries was calculated based on the mass of ZVO. All batteries were assembled in open-air conditions. These cells were tested on a Neware battery cycler (CT-4008T-5V10mA-164, Shenzhen, China) at room temperature.

## Characterizations

Raman spectroscopy for the electrolyte structure was conducted on Horiba LabRAM HR Evolution microscope with a 532 nm or 633 nm excitation laser; In situ Raman was performed by using a ZVO electrode as the working electrode, and Zn electrode as the reference and counter electrode. The in situ Raman cell consists of the ZVO electrode, Zn electrode, separator, and electrolytes (~150 μL) with 2032-type coin batteries. The separator, Zn foil, and positive case are all perforated. Galvanostatic charge and discharge (1 A g$^{-1}$) were carried out on LAND 2001A. In situ tests were conducted at room temperature. For SERS measurements, $SiO_2$-shell insulated Au nanoparticles (CP-2, PERSer Nanotechnology) were dispersed on the Zn surface; SEM and EDS Mapping. The micro-morphology of samples before and after cycling was characterized by SEM with an acceleration voltage of 5 kV. The micro-morphology of samples was observed by a white light interference microscopic. EDS Mapping images all sampling 5 min with excitation voltage 20 kV. SEM images were obtained using a FEI-SEM 7800 F Prime microscope (JEOL Ltd., Japan) equipped with an EDS attachment. The morphology of ZVO was characterized by TEM (JEOL-2100 F, 200 kV) with energy dispersive spectroscopy (EDS) for elemental analysis; XRD (Rigaku SmartLab 9 kW, Cu Kα, λ = 1.54056 Å) was utilized for characterizing Zn deposition. X-ray photoelectron spectroscopy (XPS, 250XI ESCALAB Thermo Fisher Scientific) was used to analyze the chemical composition of the surface of Zn electrode; In situ pH was performed by using a Zn electrode as the working electrode, and another Zn electrode as the reference and counter electrode. The in-situ pH test device is shown in Supplementary Fig. 2, we employed the MODEL 6173 pH meter (JENCO INSTRUMENTS CO., LTD.) for pH measurements, a device designed with microcomputer functionality and equipped with the STMirco5 pH electrode (φ 5 mm, Ohaus International Trading (Shanghai) Co., Ltd.) for precise pH measurements. The in-situ pH data were automatically logged by a computer system. The gaseous products were analyzed using a gas chromatograph (Fuli instrument, GC9720plus). $H_2$ was analyzed by a thermal conductivity detector (TCD). To facilitate the detection of gas production, we used a home-made battery module (Supplementary Fig. 46) for both battery cycling and gas production testing. The battery module is configured with air inlet and outlet valves for gas collection.

## Computational methods

Molecular dynamics (MD) simulations were carried out by using CP2K software with PM6-D3 method[51]. The gamma point was used for the Brillouin zone sampling. MD simulations were carried out in the canonical (NVT) ensemble using a Nosé–Hoover chain thermostat to maintain the average temperature at 320 K with the time constant of 100 fs. A time step of 1 fs was used in all MD simulations. The MD simulations were run for 20 ps to yield the data, and the last 10 ps

trajectory was used for analysis. Initial structures were constructed using the Packmol software[52]. The dimensions of the simulation are 26.2 Å, 22.7 Å, 74.4 Å in the x, y, z directions with a large vacuum region (z axis). The simulation box contains 362 $H_2O$, 7 $Zn^{2+}$, 7 $SO_4^{2-}$ in the base electrolyte and one more HTFSI in the modified electrolyte. The metal slab contains four Zn atom layers. The density statistics of systems were handled with VMD software[53].

## Reporting summary

Further information on research design is available in the Nature Portfolio Reporting Summary linked to this article.

## Data availability

The experimental data that support the findings of this study are available from the corresponding authors upon request. Source data are provided with this paper.

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

## Acknowledgements

This study was supported by the National Key R&D Program of China (2021YFA1201800), the National Natural Science Foundation of China (grant no. 22179124, 21905265), the Fundamental Research Funds for the Central Universities (WK3430000007). The authors are grateful for resources from the the Supercomputing Center, the Center for Micro and Nanoscale Research and Fabrication and the Instruments Center for Physical Science at USTC.

## Author contributions

Q.N. and X.R. conceived the idea and designed the experiments. Q.N. performed the main research work with the help of X.L., D.R., Y.L., B.-Q.X., Z.C., Z.W., Q.D., and J.F. Q.D. conducted the XRD test. Z.W. and J.J. performed the Raman test. J.M. Z.M. and D.W. carried out the SEM test. Q.N. and X.R. wrote the manuscript, and all authors discussed the results and revised the manuscript.

## Competing interests

The authors declare no competing interests.
