## [Peer Review File · Nature Communications]

Highly Reversible Zinc Metal Anode Enabled by Strong Brønsted Acid and Hydrophobic Interfacial ChemistryREVIEWER COMMENTS

Reviewer #1 (Remarks to the Author):

In this work, the authors present a “fighting fire with fire” strategy to improve the reversibility of the Zn anode in aqueous electrolyte using an extremely strong Bronsted acid. Despite its high protonic activity, HTFSI could effectively avoid the notorious alkaline corrosion side products (Zn hydroxide sulfate) to induce uniform Zn nucleation and simultaneously avoid alkaline corrosion to the cathode. To understand the reason for the highly reversible Zn anode, the authors conducted detailed mechanistic studies and revealed the unique role of the hydrophobic anion TF₂SI⁻ at the interface. This topic is highly interesting and the overall discussion is consistent. The reviewer would be happy to recommend the publication of this manuscript after minor revision. A few questions are listed below.

1. The introduction of HTFSI has also significantly improved the CE of Zn//Cu cells in the initial cycle, and it is recommended to add more discussion to this behavior.
2. I am curious whether HTFSI has any effect on the solvation structure of the electrolyte.
3. The choice of ZnSO₄ as the electrolyte salt is noted in this article. I wondered if HTFSI has a comparable effect when used with other salts.
4. It is interesting that the SEI on Zn metal is dominated by ZnSO_x and ZnS species, instead of ZnF₂ frequently found in previous studies. Could the authors add discussion about their roles in the SEI?
5. The measurement of the electrolyte ionic conductivity and the gas chromatography test are not described in the manuscript. Please add them to the experimental session.
6. It would be beneficial to include more electrochemical results, such as cyclic voltammetry (CV) testing of Zn//Zn cells and the electrochemical window of the electrolyte for a better understanding.

Reviewer #2 (Remarks to the Author):

This work reports on the use of 0.1 m HTFSI additive in ZnSO₄ electrolyte to enable the reversible operation of Zn negative electrode. Such additive leads to a decrease of the pH value, which suppresses the formation of insoluble alkaline byproducts of Zn. This strategy was called a “fighting fire with fire strategy of using an extremely strong Bronsted acid” and “a non-intuitive strategy” by the authors. At this point, one may suggest to avoid an overselling approach adopted by the authors. In addition, the mechanism behind the observed improvement is unclear: the title highlights the Bronsted acidity while, in the manuscript, the authors compared the electrochemical properties obtained with the addition of HOTf demonstrating the important role of the anion. The formation of ZnS characterized by XPS is not convincing as Ar sputtering might trigger the reduction of sulfates. Overall, a lot of characterization techniques have been performed, but not analyzed in depth (particularly the DFT part) and gathered in a superposition way. While the work seems to provide a Zn battery with enhanced properties, the manuscript is not scientifically sound and therefore is not appropriate for Nature Comm.

Reviewer #3 (Remarks to the Author):

Nian et al. have reported a compelling discovery that the incorporation of a small quantity of superacid HTFSI into the electrolyte leads to a significant enhancement in the reversibility of the Zn anode. Furthermore, the authors have shed light on the role and mechanism of HTFSI through well-designed experiments and theoretical calculations. This work contributes valuable insights that challenge conventional understanding of Zn metal in aqueous electrolytes, which has typically been regarded as unstable in acidic environments. The effective removal of the corrosion side products (on both electrodes) and the formation of a uniform SEI layer due to the HTFSI acid are proven to be critical for the greatly improved performance. This is of great importance for the relevant research field.

Therefore, this article is suitable for publication in Nature Communications. I also have some questions regarding this article, which are described as following.

-It is difficult to gain a detailed understanding of the in-situ pH device based on the current descriptions. Therefore, the author should furnish comprehensive information about the apparatus used in the manuscript, including electrode dimensions, instrument models, and, if possible, optical photographs of the device for reference.

-In Fig. 2e, Zn foil exhibits noticeable byproducts after being immersed in a 1 m ZnSO₄ electrolyte for 2 hours. It is better to conduct replicative experiments to validate this result. Additionally, a comparison with pristine Zn foil should be included.

-In Fig. 3a, the use of a 1 m ZnSO₄-0.1 HTFSI electrolyte, in comparison to a 1 m ZnSO₄ electrolyte, not only enhances the reversibility of the battery but also significantly extends the cycle life of the Zn-Cu cell. It is advisable that the author carries out corresponding parallel experiments to ensure the repeatability of the results.

-It is recommended to investigate the effects of introducing other types of acids, such as H₃PO₄ and HCl, on zinc anodes. This would assist in establishing a set of guidelines to make informed choices regarding the use of different acids.

-In the section discussing 'The working mechanism of HTFSI', the author emphasizes that TFSI- adsorbs on the electrode surface. It is worth exploring whether this adsorption characteristic has an impact on the crystal plane orientation of Zn deposition.

-According to the information presented in this paper, there appears to be a synergistic effect between H⁺ and TFSI⁻. To confirm this effect, could the author provide additional evidence or experiments that validate the existence of this synergy?

Point-by-point response to reviewers' comments for NCOMMS-23-39968

We would like to thank the reviewers for their valuable comments. We have incorporated the reviewers' comments and suggestions into the revised manuscript. We also provided detailed answers and explanations to the reviewers' comments. The changes to the manuscript are marked **yellow** in this response and the revised manuscript.

Reviewer #1 (Remarks to the Author):

In this work, the authors present a “fighting fire with fire” strategy to improve the reversibility of the Zn anode in aqueous electrolyte using an extremely strong Bronsted acid. Despite its high protonic activity, HTFSI could effectively avoid the notorious alkaline corrosion side products (Zn hydroxide sulfate) to induce uniform Zn nucleation and simultaneously avoid alkaline corrosion to the cathode. To understand the reason for the highly reversible Zn anode, the authors conducted detailed mechanistic studies and revealed the unique role of the hydrophobic anion TFSI⁻ at the interface. This topic is highly interesting and the overall discussion is consistent. The reviewer would be happy to recommend the publication of this manuscript after minor revision. A few questions are listed below.

Response: We would like to thank the reviewer for the positive comments on our work. Based on your suggestions, we have included additional experiments and analyses. Our point-by-point responses to comments are detailed as follows.

1. The introduction of HTFSI has also significantly improved the CE of Zn//Cu cells in the initial cycle, and it is recommended to add more discussion to this behavior.

Response: Thanks for your kind suggestion. We appreciate your careful review of our work. We have calculated the initial CEs of Zn||Cu cells based on parallel cells. As shown in Fig. R1, the average initial CEs in 1 m ZnSO₄ and 1 m ZnSO₄+0.1 m HTFSI

are 81.91% (standard deviation of 2.79%) and 93.72% (standard deviation of 0.43%), respectively. The addition of HTFSI greatly improves the initial CE of Zn plating/stripping. This indicates that the removal of the alkaline corrosion byproducts during the rest period is highly beneficial for improving the Zn metal reversibility by controlled Zn deposition.

Fig. R1. Initial Zn CEs in Zn||Cu cells with different electrolytes at 1 mA cm^{-2} , 0.5 mAh cm^{-2} .

We have also added the above result (as Supplementary Fig. 13) and the following discussion to the revised manuscript:

Line 211-216: “In addition, the initial CEs in 1 m ZnSO₄ and 1 m ZnSO₄+0.1 m HTFSI are 86.48% and 93.27%, respectively (Supplementary Fig. 13). The addition of HTFSI greatly improves the initial CE of Zn plating/stripping. This indicates that the removal of the alkaline corrosion byproducts during the rest period is highly beneficial for improving the Zn metal reversibility by controlled Zn deposition.”

2. I am curious whether HTFSI has any effect on the solvation structure of the electrolyte.

Response: Thank you for your insightful comment regarding the potential impact of HTFSI on the solvation structure of the electrolyte. We have supplemented Raman spectroscopy experiments to investigate the effect of HTFSI on the electrolyte solvation structure. As shown in Fig. R2, following the introduction of HTFSI, the S-O vibration

signals of SO_4^{2-} and the O-H vibration signal of H_2O exhibit no apparent changes. This suggests that the addition of HTFSI has no apparent effect on the electrolyte solvation structure. This could be attributed to the low amount and highly dissociating nature of the TFSI^- anion added.

Fig. R2. The Raman spectra of the 1 m $\text{ZnSO}_4+0.1$ m HTFSI (red line) and 1 m ZnSO_4 (black line) electrolytes.

We have also added the above result (as Supplementary Fig. 30) and the following discussion to the revised manuscript:

Line 346-349: “As indicated by the Raman spectrum shown in Supplementary Fig. 30, the S-O vibration signals related to SO_4^{2-} and the O-H vibration signal associated with H_2O exhibit no apparent changes after the addition of HTFSI. This suggests that the effect of HTFSI cannot be attributed to the change in the electrolyte solvation structure.”

3. The choice of ZnSO_4 as the electrolyte salt is noted in this article. I wondered if HTFSI has a comparable effect when used with other salts.

Response: We appreciate your insightful comments about the potential effects of HTFSI with other salts. We have conducted additional experiments using other salts (ZnCl_2 and $\text{Zn}(\text{CH}_3\text{COO})_2$) in the electrolyte (Fig. R3). Upon the addition of 0.1 m HTFSI to electrolytes containing 1 m ZnCl_2 or 1 m $\text{Zn}(\text{CH}_3\text{COO})_2$, notable enhancements in both the initial Coulombic efficiency (ICE) and the average Coulombic efficiency (ACE) over the first 100 cycles of the $\text{Zn}||\text{Cu}$ cell were observed. This improvement further signifies the general effect of HTFSI on the reversibility of the Zn anode. Nevertheless, we also notice that the ICE of 93.27% in ZnSO_4 -based electrolyte is much higher than those in ZnCl_2 and $\text{Zn}(\text{CH}_3\text{COO})_2$ -based (79.22% and

82.43%, respectively). In addition, no apparent formation of ZnS could be detected by grazing incidence XRD (GIXRD) on the surface of Zn foil after soaking in 1 m ZnCl₂+0.1 m HTFSI and 1 m Zn(CH₃COO)₂+0.1 m HTFSI electrolytes (Fig. R4). In contrast, ZnS could be identified on the Zn surface in the 1 m ZnSO₄+0.1 m HTFSI system (Fig. R5). It is likely that the formation of insoluble ZnS from ZnSO₄ could help mitigate the corrosion of Zn metal by electrolyte. However, other factors of anions, including charge density, anion donor number, hydrolysis propensity etc., may also influence the reversibility of Zn anode and cannot be excluded in the present study. Further detailed studies are warranted to resolve the complicated nature of the chemical/electrochemical interfacial processes in Zn metal batteries.

We have also added the above result (as Supplementary Fig. 24) and the following discussion to the revised manuscript:

Line 270-275: “Furthermore, our tests using ZnCl₂ and Zn(CH₃COO)₂ as the Zn salts also demonstrated improved Zn CEs in Zn||Cu cells with the addition of 0.1 m HTFSI, as shown in Supplementary Fig. 24. This indicates the general positive effect of HTFSI on the Zn anode reversibility. Their differences of Zn CEs compared to that in the ZnSO₄-based electrolyte could be mainly attributed to the SEI composition, which will be discussed in the later session.”

Line 373-381: “We also employed a non-destructive characterization method, grazing incidence X-ray diffraction (GIXRD), to identify the existence of ZnS. With the incident X-ray nearly parallel to the sample surface, GIXRD is particularly suitable for characterizing the surface structure. As illustrated in Supplementary Fig. 33, the ZnS signal could be detected through GIXRD. This additional evidence further supports that ZnS is actually formed on the Zn surface, not due to Ar⁺ sputtering. However, no discernible ZnS was detected from GIXRD on the Zn surfaces immersed in 1 m ZnCl₂+0.1 m HTFSI and 1 m Zn(CH₃COO)₂+0.1 m HTFSI electrolytes (Supplementary Fig. 34). This further confirms that the formation of ZnS from SO₄²⁻.”

Fig. R3. Zn CE evolution in Zn||Cu cells with electrolytes based on different salts at 1 mA cm⁻², 0.5 mAh cm⁻².

Fig. R4. GIXRD patterns of the Zn anodes after soaking in 1 m ZnCl₂+0.1 m HTFSI (a) and 1 m Zn(CH₃COO)₂+0.1 m HTFSI (b) for 5 h.

Fig. R5. GIXRD pattern of the Zn anode after soaking in 1 m ZnSO₄+0.1 m HTFSI for 5 h.

4. It is interesting that the SEI on Zn metal is dominated by ZnSO_x and ZnS species, instead of ZnF₂ frequently found in previous studies. Could the authors add discussion about their roles in the SEI?

Response: We appreciate your insightful question and suggestion.

We have expanded the discussion of the roles of ZnS and ZnF₂ species in SEI in the revised manuscript.

The specific modifications are as follows:

Line 381-386: “While ZnF₂ was typically regarded as the favorable SEI component for the Zn anode, as an analogy to the Li anode, for inhibiting electrolyte side reactions and dendrite growth, it is noteworthy that ZnF₂ has a much higher solubility in water than that of ZnS (solubility constant $K_{sp} = 3.04 \times 10^{-2}$ vs 2.5×10^{-22} , 25 °C)^{49, 50}. Therefore, a ZnS-enriched SEI layer could potentially be more effective in isolating the active Zn from electrolyte corrosion.”

Line 725:

49 David, R. L. CRC Handbook of Chemistry and Physics, 84th Edition (2004).

Line 726:

50 John, A. Dean. Lange’s Handbook of Chemistry, 15th Edition (1999).

5. The measurement of the electrolyte ionic conductivity and the gas chromatography test are not described in the manuscript. Please add them to the experimental session.

Response: Thanks for your careful review and kind suggestion. This new version provides a comprehensive description of the methods employed for measuring electrolyte ionic conductivity and conducting the gas chromatography test. We have included details of the experimental setup (Fig. R6 and Fig. R7) and specific parameters that were considered during these analyses.

We have also added the supplementary information (as Supplementary Fig. 43-45) and the following discussion to the revised manuscript:

Line 567-571: “The gaseous products were analyzed using a gas chromatograph (Fuli instrument, GC9720plus). H₂ was analyzed by a thermal conductivity detector (TCD). To facilitate the detection of gas production, we used a home-made battery module (Supplementary Fig. 45) for both battery cycling and gas production testing. The battery module is configured with air inlet and outlet valves for gas collection. ”

Line 508-517: “The ionic conductivity measurements were conducted in a micro-electrochemical cell (electrolyte volume ~1 mL, Supplementary Fig. 43), with a two-electrode setup (titanium foils as the electrodes). The impedance of the target electrolyte was first measured using EIS (100 mHz~1 MHz, 10 mV). Then the standard 1 M KCl electrolyte was tested as a reference (0.1118 S cm⁻¹, 25 °C) to calculate the ionic conductivity of the target electrolytes. The resulting ionic conductivity data is depicted in Supplementary Fig. 44. Notably, the ionic conductivity of the 1 m ZnSO₄+0.1 m HTFSI electrolyte surpasses that of the 1 m ZnSO₄ electrolyte. This enhancement can be attributed to the increased concentration of highly conductive protons in the electrolyte due to the addition of HTFSI. ”

Fig. R6. Photo of the battery module used for gas chromatography tests.

Fig. R7. Photo of the electrochemical cell used for ionic conductivity tests.

Fig. R8. Ionic conductivity of electrolytes.

6. It would be beneficial to include more electrochemical results, such as cyclic voltammetry (CV) testing of Zn//Zn cells and the electrochemical window of the electrolyte for a better understanding.

Response: We appreciate your constructive feedback and suggestions.

We have conducted additional experiments and have included the results in the

supporting information of the revised manuscript. As illustrated in Fig. R9, the CV scan results of Zn||Zn cells in the two electrolytes exhibit noticeable distinctions, particularly in the response currents. The addition of HTFSI leads to a reduction in the response current. This likely indicates a smaller reaction surface area during the electrochemical process, which is consistent with the more uniform and dense deposition of Zn metal (*J. Am. Chem. Soc.* 25, 11129 – 11137 (2022)).

We have also added the above result (as Supplementary Fig. 18 and 19) and the discussion to the supporting information of the revised manuscript.

Fig. R9. CV curves of Zn||Zn symmetrical cells with a scan speed of 5 mV s⁻¹.

To prevent the oxidation of Cu foil during the anodic scan, we assembled Zn||Ti cells for electrochemical stability window tests. It's noteworthy that distinguishing the hydrogen evolution potential and Zn deposition potential in aqueous electrolytes containing Zn salts can be challenging. Typically, sodium or lithium salts are used instead of Zn salts in electrochemical stability window tests. However, considering that the SEI layer on the Zn surface in the 1 m ZnSO₄+0.1 m HTFSI electrolyte is indispensable for inhibiting hydrogen evolution, we did not replace the Zn salt in this case. As shown in Fig. R10, in comparison to the 1 m ZnSO₄ electrolyte, the oxygen evolution onset potential was increased in the 1 m ZnSO₄+0.1 m HTFSI electrolyte, potentially due to the hydrophobic interface induced by TFSI⁻ anions. During the cathodic scan, it is difficult to differentiate the hydrogen evolution reaction and the Zn deposition reaction. However, on closer inspection, the initial increase of the cathodic current was delayed in the ZnSO₄+0.1 m HTFSI electrolyte, which is possibly due to

the formation of the SEI.

Fig. R10. Electrochemical stability window of electrolyte with a scan speed of 5 mV s⁻¹.

Reviewer #2 (Remarks to the Author):

This work reports on the use of 0.1 m HTFSI additive in ZnSO₄ electrolyte to enable the reversible operation of Zn negative electrode. Such additive leads to a decrease of the pH value, which suppresses the formation of insoluble alkaline byproducts of Zn. This strategy was called a “fighting fire with fire strategy of using an extremely strong Bronsted acid” and “a non-intuitive strategy” by the authors. At this point, one may suggest to avoid an overselling approach adopted by the authors. In addition, the mechanism behind the observed improvement is unclear: the title highlights the Bronsted acidity while, in the manuscript, the authors compared the electrochemical properties obtained with the addition of HOTf demonstrating the important role of the anion. The formation of ZnS characterized by XPS is not convincing as Ar sputtering might trigger the reduction of sulfates. Overall, a lot of characterization techniques have been performed, but not analyzed in depth (particularly the DFT part) and gathered in a superposition way. While the work seems to provide a Zn battery with enhanced properties, the manuscript is not scientifically sound and therefore is not appropriate for Nature Comm.

Response: We appreciate your thorough review of our manuscript and your constructive comments. Your insights are invaluable in improving the quality of our work. We have carefully considered your suggestions and addressed the concerns raised in your comments.

1. We apologize for using unconventional descriptions for our strategy, which was meant as imagery metaphors to arouse the interest of readers. We acknowledge that it is critical to provide balanced and accurate representation of scientific findings. Therefore, we have made revisions in our discussions to ensure that our claims accurately reflect the experimental observations.

The specific modifications are as follows:

Line 14-17: “Here, we present a direct strategy to tackle such problems using an extremely strong Brønsted acid, bis(trifluoromethanesulfonyl)imide (HTFSI), as the electrolyte additive. This approach reforms the battery interfacial chemistry on both

electrodes.”

Line 85-87: “Here, we propose to modulate the corrosion pathways of Zn metal by adding a strong Brønsted acid, bis(trifluoromethanesulfonyl)imide (HTFSI), into the conventional aqueous electrolyte (1 m ZnSO₄).”

Line 445-447: “Instead, using the strong Brønsted acid (HTFSI) with hydrophobic moieties as the electrolyte additive for RAZBs effectively promotes the uniform deposition of Zn after removing insoluble alkaline byproducts at the Zn surface.”

Line 456-459: “This work unveils an intriguing discovery that strong and hydrophobic acids can yield beneficial effects on Zn anodes and provides valuable insights into the development of reversible Zn metal anodes.”

2. We apologize for causing the confusion about acidity and anion effect. Actually, the two parts are closely related to each other. The nature of the anion has a critical influence on the acidity (proton donating ability). As we revealed in the original manuscript, adding acids (including H₂SO₄, HOTf, HTFSI and etc.) effectively prevents the accumulation of self-corrosion byproducts on the Zn surface during battery resting periods, offering the possibility for subsequent uniform Zn deposition. The bulky TFSI⁻ anion with highly delocalized negative charge effectively increases its proton donating ability. Furthermore, TFSI⁻ anions are prone to accumulate on the electrode surface. Due to the hydrophobic nature of TFSI⁻, there is a reduced distribution of H₂O on the Zn surface to suppress Zn metal corrosion. These features of the HTFSI additive are instrumental in the significantly improved CE of the Zn anode. In the manuscript, we compare the electrochemical performance obtained with the addition of H₂SO₄ or HOTf, demonstrating the important role of anions. The results show that acids such as HTFSI/H₂SO₄/HOTf can remove alkaline byproducts to improve the CE of Zn anodes (Fig. 2c-g, Supplementary Fig. 3-9, and Supplementary Fig. 36-37 in the manuscript). However, the differences between SO₄²⁻, OTf⁻ and TFSI⁻

anions lead to differences in cycle performance (Supplementary Fig. 38 and 39 in the manuscript). Therefore, these differences caused by anions cannot be ignored.

Previous literature has compared the influence of anions on the acidity and hydrophobicity. As shown in Table R1 (*Chem. Eur. J.* 22(37), 13312 (2016)), the acidity varies greatly depending on the anion property (described in terms of Hammett acidity function values, a scaling method typically used for comparing acids stronger than sulfuric acid). HTFSI (or HNTf₂) was found to have a stronger acidity than HOTf (-19.0 vs. -14.6). It is likely that the bulky TFSI⁻ anion with highly delocalized negative charge increases the proton donating ability.

In addition, previous studies also indicate that TFSI⁻ anion has a stronger hydrophobicity than OTf⁻ anion (*Energy & Environ. Sci.* 16, 1480-1501 (2023)). Comparing the interaction energy between the anion and water molecule with that between two water molecules (-24.48 kJ/mol), the TFSI⁻ anion exhibits apparently higher hydrophobicity than the OTf⁻ anion (5.67 kJ/mol vs. -5.69 kJ/mol), while the SO₄²⁻ anion is apparently hydrophilic (~-42.71 kJ/mol) (Fig. R11). The anion's hydrophobic or hydrophilic characteristics lead to inherently distinct interfacial behaviors and Zn metal anode performances.

Table R1. Acidity comparison of different acids (*Chemistry-A European Journal* 22(37), 13312 (2016)).

Table 2. Hammett acidity function values.		
Acid	¹³ C [ppm]	H ₀
HOMs	162.5	-8.1
HCl	161.2	-11.2
H ₂ SO ₄	160.89	-12
HOTf	159.82	-14.6
FSO ₃ H	159.6	-15.1
HClO ₄	159.18	-16.1
HNTf ₂	157.96	-19.0
HBf ₄	156.42	-22.7
HSbF ₆	156.05	-23.6

Fig. R11. Water-ion interaction energy as function of the ions charge density (*Energy Environ. Sci.* 16, 1480-1501, (2023)).

We have also conducted supplementary experiments to confirm the synergistic effect between H^+ and TFSI^- . We prepared 0.9 m $\text{ZnSO}_4 + 0.1$ m $\text{Zn}(\text{TFSI})_2$ and 1 m $\text{ZnSO}_4 + 0.05$ m H_2SO_4 electrolytes (Zn^{2+} concentrations were kept the same) and studied the effects of H^+ and TFSI^- on Zn deposition, respectively. As shown in Fig. R12, the addition of TFSI^- alone does not effectively enhance battery performance. Considering the inferior cycling performance in the 1 m $\text{ZnSO}_4 + 0.05$ m H_2SO_4 electrolyte (Fig. R13), it is evident that the sole addition of TFSI^- or H^+ is not very effective. In contrast, the addition of HTFSI significantly enhances the performance of the Zn anode, indicating a synergistic effect between H^+ and TFSI^- in improving Zn anode reversibility.

Fig. R12. Zn CE evolution in Zn||Cu cells with 0.9 m ZnSO₄+0.1 m Zn(TFSI)₂ electrolytes at 1 mA cm⁻², 0.5 mAh cm⁻²; (a) and (b) are parallel battery data.

Fig. R13. Zn CE evolution in Zn||Cu cells with 1 m ZnSO₄+0.05 m H₂SO₄ electrolytes at 1 mA cm⁻², 0.5 mAh cm⁻²; (a) and (b) are parallel battery data.

We have also added the above result and the following discussion to the revised manuscript as follows:

To better emphasize the article's purpose and highlight the important role of Bronsted acidity and anions, we have revised the title to "Highly Reversible Zinc Metal Anode Enabled by Strong Brønsted Acid and Hydrophobic Interfacial Chemistry "

Line 408-411: "It should also be noted that TFSI⁻ anion alone cannot afford reversible Zn metal cycling. The inferior Zn anode performances with only H⁺ or TFSI⁻ additive (Supplementary Fig. 38 and 41) suggest the synergistic effect between H⁺ and TFSI⁻."

Line 437-441: "These two aspects are both closely related to the property of the TFSI⁻ anion, as the bulky structure with hydrophobic -CF₃ moieties not only delocalizes the negative charge to increase the proton donating ability but fosters a hydrophobic interface at the Zn surface to suppress further electrolyte corrosions."

3. ZnS Characterization: We fully understand your concern regarding the XPS characterization of ZnS. To confirm whether Ar⁺ sputtering induces the reduction of sulfate to ZnS, we conducted additional experiments by directly sputtering of the ZnSO₄ salt. As depicted in Fig. R14, after Ar⁺ sputtering for 120 s, no ZnS signal was detected in the binding energy range of 160 ~165 eV. This suggests that the presence of ZnS on the electrode surface may not be attributed to Ar⁺ sputtering.

The parameters of Ar⁺ sputtering for XPS tests are as follows :

Ion gun energy: 5 keV

Incident angle: 90°

Etching area: 2.0 mm*2.0 mm

Etching time: 120 s

Fig. R14. S 2p XPS of ZnSO₄ before and after Ar⁺ sputtering.

Furthermore, we have also employed a non-destructive detection method, grazing incidence X-ray diffraction (GIXRD), to identify the existence of ZnS. With the incident X-ray nearly parallel to the sample surface, GIXRD is particularly suitable for characterizing the surface structure. As illustrated in Fig. R15, the ZnS signal could be detected through GIXRD. This additional evidence further supports that ZnS is actually formed on the Zn surface, not due to Ar⁺ sputtering. In addition, GIXRD test results showed that no ZnS component was detected on the surface of Zn foil soaked in 1 m ZnCl₂+0.1 m HTFSI and 1 m Zn(CH₃COO)₂+0.1 m HTFSI (Fig. R16). This further confirms that the formation of ZnS from SO₄²⁻.

Fig. R15. GIXRD pattern of the Zn anode after soaking in 1 m ZnSO₄+0.1 m HTFSI for 5 h.

Fig. R16. GIXRD patterns of the Zn anodes after soaking in 1 m ZnCl₂+0.1 m HTFSI (a) and 1 m Zn(CH₃COO)₂+0.1 m HTFSI (b) for 5 h.

Line 373-381: We also employed a non-destructive characterization method, grazing incidence X-ray diffraction (GIXRD), to identify the existence of ZnS. With the incident X-ray nearly parallel to the sample surface, GIXRD is particularly suitable for characterizing the surface structure. As illustrated in Supplementary Fig. 33, the ZnS signal could be detected through GIXRD. This additional evidence further supports that ZnS is actually formed on the Zn surface, not due to Ar⁺ sputtering. However, no discernible ZnS was detected from GIXRD on the Zn surfaces immersed in 1 m ZnCl₂+0.1 m HTFSI and 1 m Zn(CH₃COO)₂+0.1 m HTFSI electrolytes (Supplementary Fig. 34). This further confirms that the formation of ZnS from SO₄²⁻.

4. Further Analysis of Results: We appreciate your thorough examination of the experimental results. Following your recommendations, we have conducted a comprehensive analysis of the existing findings, particularly the DFT part. Below, we outline some of the additional analysis contents incorporated based on your suggestions.

Line 415-422: “According to DFT calculation results, SO₄²⁻ and TFSI⁻ have favorable adsorption capabilities on the Zn surface. The co-adsorption of SO₄²⁻ and TFSI⁻ on the Zn surface also facilitates the formation of ZnS. In addition, through the comparison of different crystal planes, SO₄²⁻ (-4.9124 eV (002), -5.2624 eV (101)) and TFSI⁻ (-2.6186 eV (002), -2.7702 eV (101)) potentially prefer adsorptions on the (101) plane over the (002) plane, which may account for the slight growth orientation along the [002] direction (XRD result shown in Supplementary Fig. 42).”

Line 381-386: “While ZnF₂ was typically regarded as the favorable SEI component for the Zn anode, as an analogy to the Li anode, for inhibiting electrolyte side reactions and dendrite growth, it is noteworthy that ZnF₂ has a much higher solubility in water than that of ZnS (solubility constant $K_{sp} = 3.04 \times 10^{-2}$ vs 2.5×10^{-22} , 25 °C)^{49, 50}. Therefore, a ZnS-enriched SEI layer could potentially be more effective in isolating the active Zn from electrolyte corrosion.”

Line 270-275: “Furthermore, our tests using ZnCl_2 and $\text{Zn}(\text{CH}_3\text{COO})_2$ as the Zn salts also demonstrated improved Zn CEs in Zn||Cu cells with the addition of 0.1 m HTFSI, as shown in Supplementary Fig. 24. This indicates the general positive effect of HTFSI on the Zn anode reversibility. Their differences of Zn CEs compared to that in the ZnSO_4 -based electrolyte could be mainly attributed to the SEI composition, which will be discussed in the later session.”

Line 405-411: “As illustrated in Supplementary Fig. 40, the test results with other acids (HCl and H_3PO_4) indicate that a suitable amount of acids can enhance the Zn CE in Zn||Cu cells. Their differences with HTFSI are likely due to the properties of anions. However, it should also be noted that TFSI^- anion alone cannot afford reversible Zn metal cycling. The inferior Zn anode performances with only H^+ or TFSI^- additive (Supplementary Fig. 38 and 41) suggest the synergistic effect between H^+ and TFSI^- .”

Reviewer #3 (Remarks to the Author):

Nian et al. have reported a compelling discovery that the incorporation of a small quantity of superacid HTFSI into the electrolyte leads to a significant enhancement in the reversibility of the Zn anode. Furthermore, the authors have shed light on the role and mechanism of HTFSI through well-designed experiments and theoretical calculations. This work contributes valuable insights that challenge conventional understanding of Zn metal in aqueous electrolytes, which has typically been regarded as unstable in acidic environments. The effective removal of the corrosion side products (on both electrodes) and the formation of a uniform SEI layer due to the HTFSI acid are proven to be critical for the greatly improved performance. This is of great importance for the relevant research field. Therefore, this article is suitable for publication in Nature Communications. I also have some questions regarding this article, which are described as following.

Response: We appreciate your positive comments on our work. Based on your suggestions, we have included additional experiments and analyses.

Our point-by-point responses to comments are detailed as follows.

-It is difficult to gain a detailed understanding of the in-situ pH device based on the current descriptions. Therefore, the author should furnish comprehensive information about the apparatus used in the manuscript, including electrode dimensions, instrument models, and, if possible, optical photographs of the device for reference.

Response: We appreciate the reviewer's kind suggestion. We have revised the experimental session to add necessary information related to the in-situ pH device in the manuscript as follows.

Line 562-567: The in-situ pH test device is shown in Supplementary Fig. 2, we employed the MODEL 6173 pH meter (JENCO INSTRUMENTS CO., LTD.) for pH measurements, a device designed with microcomputer functionality and equipped with the STMirco5 pH electrode (ϕ 5 mm, Ohaus International Trading (Shanghai) Co., Ltd.) for precise pH measurements. The in-situ pH data were automatically logged by a

computer system.

Fig. R17. Photo of the pH meter.

-In Fig. 2e, Zn foil exhibits noticeable byproducts after being immersed in a 1 m ZnSO₄ electrolyte for 2 hours. It is better to conduct replicative experiments to validate this result. Additionally, a comparison with pristine Zn foil should be included.

Response: Thanks for your careful review and kind suggestion. We have conducted parallel experiments to validate the observed byproducts on the Zn foil after soaking in the 1 m ZnSO₄ electrolyte for 2 hours (Fig. R18 and Fig. R19).

To remove the pristine oxide layer on the surface, Zn foil was sanded before use. Different areas of the polished Zn foil are shown in Fig. R18. From the EDS mapping result, the surface of the polished Zn foil shows mainly Zn signal (Fig. R19). After the Zn foil was soaked in 1 m ZnSO₄ for 2 h, the electrode surface changed significantly. A large number of flaky particles were observed on the surface. The EDS elemental mapping shows the apparent formation of S and O species on the Zn surface, which agrees with the alkaline corrosion byproducts. As indicated by the changes in atomic ratios in Fig. R19, the corrosion of Zn foil deteriorates over time. Therefore, we can confirm that Zn foil has obvious self-corrosion reactions in 1 m ZnSO₄ electrolyte to accumulate a significant amount of byproducts over the Zn surface.

Fig. R18. SEM images and EDS elemental mapping of Zn foils.

Fig. R19. The corresponding atom ratios in the EDS element mapping.

-In Fig. 3a, the use of a 1 m ZnSO₄-0.1 HTFSI electrolyte, in comparison to a 1 m ZnSO₄ electrolyte, not only enhances the reversibility of the battery but also

significantly extends the cycle life of the Zn-Cu cell. It is advisable that the author carries out corresponding parallel experiments to ensure the repeatability of the results.

Response: We appreciate your suggestions. To validate and ensure the repeatability of the observed enhancements in reversibility and cycle life with the 1 m ZnSO₄-0.1 HTFSI electrolyte, we have conducted parallel experiments (Fig. R20). The results of these additional experiments have been carefully analyzed and are presented in the revised manuscript alongside the original findings for comparison.

Line 207-210: “As depicted in Fig. 3a and Supplementary Fig. 12, the CE of the Zn electrode in 1 m ZnSO₄+0.1 m HTFSI electrolytes exhibits rapid stabilization, reaching 99% within the initial 40 cycles. Subsequently, it achieved a CE of 99.7% over 1400 cycles.”

Fig. R20. Zn CE evolution in Zn||Cu cells with 1 m ZnSO₄+0.1 m HTFSI electrolytes at 1 mA cm⁻², 0.5 mAh cm⁻².

-It is recommended to investigate the effects of introducing other types of acids, such as H₃PO₄ and HCl, on zinc anodes. This would assist in establishing a set of guidelines to make informed choices regarding the use of different acids.

Response: Thanks for your kind suggestion. We have conducted supplementary experiments to explore the effects of introducing HCl and H₃PO₄ on Zn anodes. As illustrated in Fig. R21, the addition of other acids (HCl and H₃PO₄) proves beneficial

in enhancing the initial Coulombic Efficiency (CE) of the Zn||Cu cells. Specifically, the addition of 0.1 m HCl to the 1 m ZnSO₄ electrolyte results in an initial CE of 91.23% and an average CE of 99.53% (the first 1100 cycles). Similarly, when 0.1 m H₃PO₄ is introduced, the initial CE is elevated to 91.64% with an average CE of 99.48% (the first 170 cycles), albeit a short cycle life.

We have also added the above result (as Supplementary Fig. 40) and the following discussion to the revised manuscript as follows:

Line 405-408: “As illustrated in Supplementary Fig. 40, the test results with other acids (HCl and H₃PO₄) indicate that a suitable amount of acids can enhance the Zn CE in Zn||Cu cells. Their differences with HTFSI are likely due to the properties of anions.”

Fig. R21. Zn CE evolution in Zn||Cu cells with 1 m ZnSO₄+0.1 m HCl (a) and 1 m ZnSO₄+0.1 m H₃PO₄ (b) electrolytes at 1 mA cm⁻², 0.5 mAh cm⁻².

-In the section discussing 'The working mechanism of HTFSI', the author emphasizes that TFSI adsorbs on the electrode surface. It is worth exploring whether this adsorption characteristic has an impact on the crystal plane orientation of Zn deposition.

Response: Thanks for your kind suggestion. We have conducted additional analysis to explore the potential impact of TFSI adsorption on the crystal plane orientation of Zn deposition (Fig. R22). We deposited a total amount of 10 mAh cm⁻² of Zn in different electrolytes on the Cu substrate under 1 mA cm⁻². Their XRD results are shown in Fig. R19, the higher I₀₀₂/ I₁₀₀ ratio in the 1 m ZnSO₄+0.1 m HTFSI electrolyte compared to

that in the 1 m ZnSO₄ electrolyte (1.34 vs. 1.18) indicates that a slight orientation of Zn deposition along the [002] direction, which agrees with the DFT result of anion adsorption.

We have also added the above result (as Supplementary Fig. 41) and the following discussion to the revised manuscript as follows:

Line 415-422: “According to DFT calculation results, SO₄²⁻ and TFSI⁻ have favorable adsorption capabilities on the Zn surface. The co-adsorption of SO₄²⁻ and TFSI⁻ on the Zn surface also facilitates the formation of ZnS. In addition, through the comparison of different crystal planes, SO₄²⁻ (-4.9124 eV (002), -5.2624 eV (101)) and TFSI⁻ (-2.6186 eV (002), -2.7702 eV (101)) potentially prefer adsorptions on the (101) plane over the (002) plane, which may account for the slight growth orientation along the [002] direction (XRD result shown in Supplementary Fig. 42).”

Fig. R22. XRD patterns of Zn deposited on the Cu substrate in different electrolytes (1 mA cm⁻², 10 mAh cm⁻²).

-According to the information presented in this paper, there appears to be a synergistic effect between H⁺ and TFSI⁻. To confirm this effect, could the author provide additional

evidence or experiments that validate the existence of this synergy?

Response: We appreciate your insightful question and suggestion. We have conducted supplementary experiments specifically to confirm the synergistic effect between H^+ and $TFSI^-$.

We prepared 0.9 m $ZnSO_4$ +0.1 m $Zn(TFSI)_2$ and 1 m $ZnSO_4$ + 0.05 m H_2SO_4 electrolytes (Zn^{2+} concentrations were kept the same) and studied the effects of H^+ and $TFSI^-$ on Zn deposition, respectively. As shown in Fig. R23, the addition of $TFSI^-$ alone does not effectively enhance battery performance. Considering the observed inferior cycling performance in the 1 m $ZnSO_4$ + 0.05 m H_2SO_4 electrolyte (Fig. R24), it is evident that the sole addition of $TFSI^-$ or H^+ is not very effective. In contrast, the addition of HTFSI significantly enhances the performance of the Zn anode, indicating a synergistic effect between H^+ and $TFSI^-$ in improving Zn anode reversibility.

Fig. R23. Zn CE evolution in Zn||Cu cells with 0.9 m $ZnSO_4$ +0.1 m $Zn(TFSI)_2$ electrolytes at 1 mA cm^{-2} , 0.5 mAh cm^{-2} ; (a) and (b) are parallel battery data.

Fig. R24. Zn CE evolution in Zn||Cu cells with 1 m ZnSO₄+0.05 m H₂SO₄ electrolytes at 1 mA cm⁻², 0.5 mAh cm⁻²; (a) and (b) are parallel battery data.

Line 408-411: “However, it should also be noted that TFSI⁻ anion alone cannot afford reversible Zn metal cycling. The inferior Zn anode performances with only H⁺ or TFSI⁻ additive (Supplementary Fig. 38 and 41) suggest the synergistic effect between H⁺ and TFSI⁻.”

REVIEWER COMMENTS

Reviewer #1 (Remarks to the Author):

I suggest the publication of this paper after the modifications.

Reviewer #3 (Remarks to the Author):

Happy with the response from the author, I think it can be accepted.

Reviewer #4 (Remarks to the Author):

I've been requested to consider especially the strength of the supporting modeling work in "Highly Reversible Zinc Metal Anode Enabled by Strong Brønsted Acid and Hydrophobic Interfacial Chemistry".

It's not clear to me that HTFSI isn't acting similarly to other well-known acid treatments which etch the oxide surface. In Supp Fig 8 (far right plot), it's clear the HTFSI has the same etchant effect as HCl and other acid treatments by reducing the presence of oxide species. The lower activation energy from Supp Fig 15 when combined with Supp Fig 8 suggests the improved kinetics is possibly the result of a cleaner initial surface.

In connection with the previous point, the authors show pH measurements only after soaking (up to 3 hours) and not after cycling or even up to the 10 hours soaking when it's evident from Supp Fig 7d that some chemistry has occurred. I'd point the authors to "Highly reversible Zn anode with a practical areal capacity enabled by a sustainable electrolyte and superacid interfacial chemistry" where Li et al. (the authors cite this article as #42) demonstrate that the pH increases with cycling (Supp Fig 17 in that reference). The consumption of H⁺ at rest (after soaking and after several formation cycles) and during cycling should be explored in greater detail. A big part of the authors' claim here is that the effect of HTFSI (viz., the pH remains low) is sustained to prevent accumulation of ZSH products which would distinguish it from the etchant pre-treatments mentioned above.

Related to the DFT calculations, the authors claim: "The co-adsorption of SO₄²⁻ and TFSI⁻ on the Zn surface also facilitates the formation of ZnS." This claim is not substantiated. I would argue the reduction of sulfates is likely due to HER and reduction induced by hydrogen species. This is a known pathway, <https://doi.org/10.1139/v76-524> (shown with TGA). The hydrogen species are consumed in this process to regenerate water in addition to promoting ZnS film formation which maybe accounts for reduced gas generation in Supp Fig 23. I think the authors have adequately addressed previous reviewer comments about the presence of ZnS, I just don't think TFSI is involved.

Not a lot of attention is paid to Supp Fig 11 where the authors explore a range of HTFSI concentrations. It would be an interesting point to address the effects of adding too little acid (we can already guess this, would be similar to the baseline electrolyte) vs adding too much (does this just exacerbate HER to the point where SEI can't contain it?).

The authors ascribe differences in the CEs for different acid additives to the anion. Specifically, they write, "As illustrated in Supplementary Fig. 40, the test results with other acids (HCl and H₃PO₄) indicate that a suitable amount of acids can enhance the Zn CE in Zn||Cu cells. Their differences with HTFSI are likely due to the properties of anions." I think it would be prudent to consider pH differences between the different electrolytes considering the above comments. These aren't reported in the main text or SI. Does this offer a stronger explanation for the observed trends in CEs?

There are not enough details on the VASP or CP2K calculations to reproduce the authors' results. See

below for information the authors need to include:

VASP:

VASP software version and version of the pseudopotentials used, including suffixes such as `'_sv'` or `'_GW'`. Alternatively, if pymatgen or similar were used, a named set may be given instead. For example, pymatgen has a set called MPRelaxSet which generates the input file used for all structures in the Materials Project database. Any changes from the default set should be communicated as well.

What DFT functional is used? Is dispersion included or not? If so, what dispersion model?

Was LDIPOL and the direction set?

What are the settings in the KPOINTS file? Gamma/Monkhorst-Pack? Number of k-points?

How large is the Zn (101) or Zn (002) interface? The interface width translates to an effective adsorption density on the surface.

How many layers thick is the metal slab?

Do these calculations correspond to gas phase adsorption energies or was VASPSol used? Either is fine, just make note of which.

CP2K:

What are the dimensions of the simulation? How many of each species is being simulated? How thick is the metal slab? Is the electrolyte in contact with both sides of the slab or did you use a separator of some kind (e.g., large vacuum region or layer of helium atoms)?

Is PM6-D3 able to reproduce basic properties of higher-level calculations of TFSI? In my experience with another semiempirical method, xTB GFN-2, it tends to predict TFSI-based solvate structures that are not consistent with GGAs or better functionals. This should be demonstrated to give confidence in the selection of a semiempirical model for this calculation.

Do the authors use a force field to simulate the liquid structure before jumping to PM6-D3? Minimize the geometry with PM6-D3? Or just use the Packmol structure as is?

A thermostat is given but not a temperature.

In the text, authors claim Figure 6e is a VDOS but it appears to just be a number density vs distance from the Zn surface.

The observation from AIMD that TFSI is preferentially adsorbed to the exclusion of sulfate is counter to the claims made by the adsorption energy calculations and ZnS formation by SO₄ reduction.

More generally, AIMD simulations often run the risk of authors jumping to conclusions with too little data to support them. I suspect the authors have done so here: double layer structure is inferred from a single trajectory with a single TFSI species from a 20 ps simulation. This is not enough time to generate a converged statistical description of the double layer structure. Given that the result also contradicts other more substantiated parts of the paper, I would suggest removing the AIMD altogether.

Overall, I think this is an intriguing work though it is largely inspired by Li et al (reference #42 here). However, I think there are still some open questions about the mechanism that need answering as well as the authors should add the necessary information to replicate the simulation results. I would

suggest the authors remove the AIMD simulation results to better align the remaining DFT results with the experimental observations.

Point-by-point response to reviewers' comments for NCOMMS-23-39968A

We would like to thank the reviewers for their valuable comments. We have incorporated the reviewers' comments and suggestions into the revised manuscript. We also provided detailed answers and explanations to the reviewers' comments. The changes to the manuscript are marked **yellow** in this response and the revised manuscript.

REVIEWER COMMENTS

Reviewer #1 (Remarks to the Author):

I suggest the publication of this paper after the modifications.

Response: We truly appreciate the reviewer's comments and suggestions to improve the quality of our manuscript.

Reviewer #3 (Remarks to the Author):

Happy with the response from the author, I think it can be accepted.

Response: We truly appreciate the reviewer's comments and suggestions for improving our manuscript.

Reviewer #4 (Remarks to the Author):

I've been requested to consider especially the strength of the supporting modeling work in "Highly Reversible Zinc Metal Anode Enabled by Strong Brønsted Acid and Hydrophobic Interfacial Chemistry".

Response: We would like to thank the reviewer for evaluating our work. Based on your suggestions, we have included additional experiments and analysis. Our point-by-point responses to comments are detailed as follows.

1. It's not clear to me that HTFSI isn't acting similarly to other well-known acid treatments which etch the oxide surface. In Supp Fig 8 (far right plot), it's clear the HTFSI has the same etchant effect as HCl and other acid treatments by reducing the

presence of oxide species. The lower activation energy from Supp Fig 15 when combined with Supp Fig 8 suggests the improved kinetics is possibly the result of a cleaner initial surface.

Response: We thank the reviewer for the question. To better articulate the uniqueness of our study, we assembled Zn//Cu cells to compare the differences between the HTFSI additive strategy and the conventional acid pre-treatment method. As shown in Figure R1, the Zn foils treated with acids (clean the Zn foil with 0.1 m acids for ten minutes, then wash the Zn foil three times with acetone and water) exhibit lower CEs and faster deterioration, contrasting with the HTFSI additive strategy. The possible reason is that, although acid pre-treatment can remove the surface oxide layer of the electrode, once Zn comes into contact with the electrolyte, corrosion by-products may still adhere to the electrode surface, affecting the uniform deposition of Zn.

Figure R1. Comparison of Zn deposition/stripping properties of acid-pretreated Zn foil and in the electrolyte with 0.1 m HTFSI additive.

We have added the above result (as Supplementary Fig. 21) and the following discussion to the revised manuscript:

line 245-249: “In addition, we also compared the differences between traditional acid pre-treatment methods and acid additive strategies. As shown in Supplementary Fig. 20, compared with the traditional acid pre-treatment method of Zn foil, the addition of HTFSI in the electrolyte greatly improves the CE of Zn plating/stripping during long-term cycling.”

2. In connection with the previous point, the authors show pH measurements only

after soaking (up to 3 hours) and not after cycling or even up to the 10 hours soaking when it's evident from Supp Fig 7d that some chemistry has occurred. I'd point the authors to "Highly reversible Zn anode with a practical areal capacity enabled by a sustainable electrolyte and superacid interfacial chemistry" where Li et al. (the authors cite this article as #42) demonstrate that the pH increases with cycling (Supp Fig 17 in that reference). The consumption of H⁺ at rest (after soaking and after several formation cycles) and during cycling should be explored in greater detail. A big part of the authors' claim here is that the effect of HTFSI (viz., the pH remains low) is sustained to prevent accumulation of ZSH products which would distinguish it from the etchant pre-treatments mentioned above.

Response: We are grateful for the valuable feedback from the reviewer.

As shown in Figure R2a, we extended the rest time of the Zn/1 m ZnSO₄+0.1 m HTFSI/Zn cell and observed that the pH increased to 3.58 after a 10-hour rest, indicating the consumption of H⁺ during the rest period. The reason for this may be insufficient formation of the solid electrolyte interphase (SEI) during the rest process, leading to limited protective effects. Although the acid has a high efficiency to remove Zn surface oxides, prolonged resting may adversely affect the stability of Zn.

In addition, we carried out additional experiments to compare the evolution of the pH after long-term resting and battery cycling (initial rest for 2 h before cycling) for the same time. In Figure R2b, after the first two cycles, the pH value is nearly the same to that after resting for the same time. However, with longer cycling, the pH value is lower than that after resting for the same time. The Zn/1 m ZnSO₄+0.1 m HTFSI/Zn cell maintains a relatively low pH during the cycling process, suggesting that the formation of a more uniform SEI during the electrochemical process provides better protective effects. The greatly improved protection ability of the SEI layer formed in the 1 m ZnSO₄+0.1 m HTFSI electrolyte is crucial for the long-term battery cycling stability.

Figure R2 a) Interfacial pH evolution of Zn/Zn symmetrical cell after resting for 0-10 h; b) pH comparison during resting and cycling.

We have added the above result (as Supplementary Fig. 3) and the following discussion to the revised manuscript:

line 156-162: “As shown in Supplementary Fig. 3a, we extended the rest time of the Zn/1 m ZnSO₄+0.1 m HTFSI/Zn cell and observed that the pH increased to 3.58 after a 10-hour rest. However, with longer cycling, the pH value is lower than that after resting for the same time (Supplementary Fig. 3b). The Zn/1 m ZnSO₄+0.1 m HTFSI/Zn cell maintains a relatively low pH during the cycling process, suggesting that the formation of a more uniform and protective SEI during the electrochemical process.”

line 155-156: this indicates that the addition of HTFSI can maintain a relatively low pH at the Zn/electrolyte interface.

3. Related to the DFT calculations, the authors claim: “The co-adsorption of SO₄²⁻ and TFSI⁻ on the Zn surface also facilitates the formation of ZnS.” This claim is not substantiated. I would argue the reduction of sulfates is likely due to HER and reduction induced by hydrogen species. This is a known pathway, <https://doi.org/10.1139/v76-524> (shown with TGA). The hydrogen species are consumed in this process to regenerate water in addition to promoting ZnS film formation which maybe accounts for reduced gas generation in Supp Fig 23. I think the authors have adequately addressed previous reviewer comments about the presence of ZnS, I just don't think

TFSI is involved.

Response: We appreciate your insightful question and suggestion.

We apologize for any confusion caused by our description of the DFT results. The information provided by the reviewers has important reference value. In the literature, it is mentioned that H₂ can reduce ZnSO₄ to form ZnS at high temperatures (>400 °C). However, we did not observe the formation of reduced products at room temperature (e.g., insoluble ZnS) when H₂ was purged through the ZnSO₄ electrolyte for 2 hours (Fig. R3). The ability of H₂ to reduce ZnSO₄ at high temperatures may not necessarily represent the same reaction occurring at room temperature. Considering the differences between chemical and electrochemical processes and taking into account the findings in the work by *Joule* 6, 1103–1120 (2022), we cannot exclude the possibility of the formation of ZnS due to reactive H species. Therefore, to avoid misunderstandings, we have removed the discussion of DFT part (further explanation in the latter session).

Simultaneously, we agree that TFSI⁻ may not directly participate in the formation of ZnS, but the role of HTFSI should not be overlooked. When the pH>5, alkaline by-products may form (Chem. Cent. J. 5,73 (2011)), potentially competitively affecting ZnS formation. The introduction of HTFSI maintains the electrolyte at a relatively low pH, which is beneficial for avoiding the formation of alkaline by-products and stabilizing ZnS. Additionally, we have verified that the synergistic effect of H⁺ and TFSI⁻ is crucial for enhancing electrochemical performance, as shown in Fig. R4, the addition of TFSI⁻ alone does not effectively enhance battery performance. Considering the inferior cycling performance in the 1 m ZnSO₄ + 0.05 m H₂SO₄ electrolyte (Fig. R5), it is evident that the sole addition of TFSI⁻ or H⁺ is not very effective. In contrast, the addition of HTFSI significantly enhances the performance of the Zn anode, indicating a synergistic effect between H⁺ and TFSI⁻ in improving Zn anode reversibility.

Figure R3 Optical photos before and after purging H_2 through the ZnSO_4 electrolyte at room temperature.

Figure R4. Zn CE evolution in $\text{Zn}||\text{Cu}$ cells with $0.9 \text{ m ZnSO}_4+0.1 \text{ m Zn(TFSI)}_2$ electrolytes at 1 mA cm^{-2} , 0.5 mAh cm^{-2} ; (a) and (b) are parallel battery data.

Figure R5. Zn CE evolution in Zn||Cu cells with 1 m ZnSO₄+0.05 m H₂SO₄ electrolytes at 1 mA cm⁻², 0.5 mAh cm⁻²; (a) and (b) are parallel battery data.

line 440-441: As a result of the strong hydrophobicity of TFSI⁻, a H₂O-deficient region was formed on the Zn surface with greatly suppressed corrosion reactions.

4. Not a lot of attention is paid to Supp Fig 11 where the authors explore a range of HTFSI concentrations. It would be an interesting point to address the effects of adding too little acid (we can already guess this, would be similar to the baseline electrolyte) vs adding too much (does this just exacerbate HER to the point where SEI can't contain it?).

Response: Thank you very much for your valuable comments.

The stability of ZnS is closely related to the pH of the electrolyte. Studies have shown that ZnS can remain stable within the pH range of 1-11 (Int. J. Sci. Res. Public., 6, 3 (2013); Mater. Lett. 61, 4267– 4271(2007); Chem. Cent. J. 5,73 (2011)). When the proton concentration is too high, the stability of ZnS may be affected. Additionally, an excessively high proton concentration can spontaneously lead to intense hydrogen evolution reactions on the Zn surface and corrosion of Zn. These factors can adversely affect the stable adhesion of ZnS to the Zn surface. Therefore, an elevated proton

concentration significantly deteriorates the deposition and adhesion of Zn, impairing the performance of the cathode electrode. These can be well reflected from the existing experimental results (Supplementary Fig. 12 in manuscript).

line 211-213: Adding too little HTFSI will have a limited effect while adding too much HTFSI will induce uncontrollable side reactions such as HER and Zn corrosion.

5. The authors ascribe differences in the CEs for different acid additives to the anion. Specifically, they write, “As illustrated in Supplementary Fig. 40, the test results with other acids (HCl and H₃PO₄) indicate that a suitable amount of acids can enhance the Zn CE in Zn||Cu cells. Their differences with HTFSI are likely due to the properties of anions.” I think it would be prudent to consider pH differences between the different electrolytes considering the above comments. These aren’t reported in the main text or SI. Does this offer a stronger explanation for the observed trends in CEs?

Response: Thank you for your valuable feedback.

To arrive at more cautious conclusions, we adjusted the pH of different electrolytes to a consistent value of 1.57 using H₂SO₄ (the pH of 1 m ZnSO₄+0.1 m HTFSI electrolyte is 1.57). After unifying the pH, we assembled Zn//Cu half-cells with the electrolytes, as shown in Figure R6. Despite this adjustment, the HTFSI-based electrolyte (Fig. 3a and Supplementary Fig. 13 in manuscript) continued to exhibit better electrochemical performance. The underlying reasons for these differences in results may still be associated with the nature of the anions.

Figure R6 Performance comparison of different acid additions after unifying the pH value (with parallel data).

6. There are not enough details on the VASP or CP2K calculations to reproduce the authors' results. See below for information the authors need to include: VASP:

VASP software version and version of the pseudopotentials used, including suffixes such as 'sv' or 'GW'. Alternatively, if pymatgen or similar were used, a named set may be given instead. For example, pymatgen has a set called MPRelaxSet which generates the input file used for all structures in the Materials Project database. Any changes from the default set should be communicated as well.

Response: Thank you for your valuable suggestions.

The version of VASP software was 5.4.4 and the pseudopotentials were PAW_PBE type.

We have incorporated these details into our revised manuscript.

line 583-584: The adsorption structures were optimized by using VASP5.4.4 package⁵¹ and the pseudopotentials were PAW_PBE type.

What DFT functional is used? Is dispersion included or not? If so, what dispersion model?

Response: Thank you for your insightful feedback. In our study, the DFT function was GGA function. Considering the possible overcorrection caused by dispersion, dispersion correction was not considered.

We've removed descriptions of DFT. The revisions to the DFT are not included in the revised manuscript.

Was LDIPOL and the direction set?

Response: Thank you for your meticulous review and your query regarding LDIPOL and the specified direction set. In our calculations, dipole correction was used and the direction was Z axis, which was vertical to the Zn slab.

We will ensure to explicitly state the LDIPOL method's usage and the corresponding direction set in our revised manuscript.

line 586-587: Dipole correction was used and the direction was Z axis, which was vertical to the Zn slab.

What are the settings in the KPOINTS file? Gamma/Monkhorst-Pack? Number of k-points?

Response: Thank you for your thoughtful review and your inquiry regarding the settings in the KPOINTS file. In our simulations, we employed the Monkhorst-Pack scheme for generating k-points, with a 2x2x1 grid. The choice of the Monkhorst-Pack scheme ensures an accurate sampling of the Brillouin zone, and the specific dimensions were determined to balance computational efficiency with convergence accuracy.

We have included this information in the revised manuscript.

line 588-589: The Monkhorst-Pack type was used and the number of k-points was 2x2x1.

How large is the Zn (101) or Zn (002) interface? The interface width translates to an effective adsorption density on the surface.

Response: Thank you for your insightful feedback and your query regarding the dimensions of the Zn (101) or Zn (002) interface in our study.

In our simulations, the dimensions of the Zn (101) or Zn (002) interface were 13.1 Å in length and 13.1 Å in width with an adsorbed species on the surface. These dimensions were chosen to accurately capture the interface properties while maintaining computational feasibility.

We ensured that interface dimensions were explicitly mentioned in our revised manuscript.

line 587-588: The size of the interface is 13.1 Å in length and 13.1 Å in width with an adsorbed species on the surface.

How many layers thick is the metal slab?

Response: We appreciate your attention to this detail. In our simulations, the metal slab consisted of four layers. This choice was made to strike a balance between capturing the relevant surface properties and computational efficiency.

We have ensured that the number of metal plate layers is explicitly stated in the revised

manuscript to provide a comprehensive description of our simulation setup.

line 584: There are four layers in the metal slab

Do these calculations correspond to gas phase adsorption energies or was VASPSol used? Either is fine, just make note of which.

Response: Thank you for your valuable input.

The calculations were corresponded to gas phase adsorption energies. VASPSol was not employed in our simulations.

We have made it clear in the revised manuscript that the reported results are based on gas phase adsorption energies.

line 592-593: The calculations were corresponded to gas phase adsorption energies.

CP2K:

What are the dimensions of the simulation? How many of each species is being simulated? How thick is the metal slab? Is the electrolyte in contact with both sides of the slab or did you use a separator of some kind (e.g., large vacuum region or layer of helium atoms)?

Response: Thank you for your thorough review and your insightful questions regarding the simulation details. In our study, the simulation dimensions were 26.2Å, 22.7Å, 74.4Å in the x, y, z direction with a large vacuum region (z axis). We simulated box containing 362 H₂O, 7 Zn²⁺, 7 SO₄²⁻ in the base electrolyte and one more HTFSI in the modified electrolyte (to represent the molar ratios in the studied electrolytes).

The metal slab in our simulations was four layers thick. Regarding the electrolyte configuration, we maintained a realistic representation by allowing the electrolyte to be in contact with both sides of the metal slab. No additional separator was used.

line 600-603: The dimensions of the simulation are 26.2Å, 22.7Å, 74.4Å in the x, y, z directions with a large vacuum region (z axis). The simulation box contains 362 H₂O, 7 Zn²⁺, 7 SO₄²⁻ in the base electrolyte and one more HTFSI in the modified electrolyte. The metal slab contains four Zn atom layers.

Is PM6-D3 able to reproduce basic properties of higher-level calculations of TFSI? In my experience with another semiempirical method, xTB GFN-2, it tends to predict TFSI-based solvate structures that are not consistent with GGAs or better functionals. This should be demonstrated to give confidence in the selection of a semiempirical model for this calculation.

Response: Thank you for raising an important point regarding the reliability of PM6-D3 in reproducing basic properties compared to higher-level calculations.

It's worth noting that PM6-D3 has proven to be a suitable choice among semiempirical methods and is known for its proficiency in simulating organic systems. To ensure the accuracy of our results, the structures of the relevant species in the electrolyte were initially optimized using quantum software before initiating the AIMD simulation.

Do the authors use a force field to simulate the liquid structure before jumping to PM6-D3? Minimize the geometry with PM6-D3? Or just use the Packmol structure as is?

Response: In our study, we opted to use the PACKMOL-generated structure directly. To ensure a realistic representation of the system, we allowed the simulation to progress for the first 10 ps, considering this period as the equilibrium stage. This approach aimed to capture the natural evolution and stabilization of the system over a short initial timeframe.

A thermostat is given but not a temperature.

Response: Thank you for your valuable input.

In our simulations, we employed the Nosé–Hoover chain thermostat to maintain the average temperature at 320 K.

line 595-598: MD simulations were carried out in the canonical (NVT) ensemble using a Nosé–Hoover chain thermostat to maintain the average temperature at 320 K with the time constant of 100 fs. A time step of 1 fs was used in all MD simulations.

In the text, authors claim Figure 6e is a VDOS but it appears to just be a number density vs distance from the Zn surface.

Response: Thank you for your careful review and valuable feedback.

Indeed, you are correct. Figure 6e does not represent a VDOS (Vibrational Density of States) as stated in the text. Rather, it depicts a number density distribution versus the distance from the Zn surface.

We sincerely apologize for any inconvenience caused by this oversight and will promptly correct the description in the revised manuscript.

line 433: The number density distribution of H₂O, SO₄²⁻, and TFSI⁻ at different distances from the Zn surface are shown in Figure 6e.

The observation from AIMD that TFSI is preferentially adsorbed to the exclusion of sulfate is counter to the claims made by the adsorption energy calculations and ZnS formation by SO₄ reduction.

Response: Thank you for your valuable feedback.

The calculation results of adsorption energy in the gas phase suggest a propensity for these species to adsorb on the surface, which is typically used to evaluate their interaction with the substrate. However, the water solvent environment is critical for the interfacial interactions in actual batteries. Therefore, we carried out AIMD simulations reveal the influence of the water solvent on the dynamics of TFSI⁻ at the interface. The conclusion derived from AIMD observations is not in direct conflict with the adsorption energy calculations and ZnS formation, but rather reflects the dynamic interplay of these factors in the realistic electrolyte environment. Our further experimental results also support the conclusion from the AIMD (shown later in the response letter). However, we agree with the reviewer that it might cause confusion for readers. The slightly improved preferential growth of Zn metal from XRD cannot be conclusively attributed to the adsorption of species on the different facets. Therefore, we have removed the DFT calculation part to avoid potential confusion.

More generally, AIMD simulations often run the risk of authors jumping to conclusions with too little data to support them. I suspect the authors have done so here: double layer structure is inferred from a single trajectory with a single TFSI species from a 20

ps simulation. This is not enough time to generate a converged statistical description of the double layer structure. Given that the result also contradicts other more substantiated parts of the paper, I would suggest removing the AIMD altogether.

Response: Thank you for your thoughtful comments and constructive feedback.

To eliminate the risk of drawing conclusions with insufficient data, we increased the number of simulations. The results of the three simulations are shown in Figure R7. Compared with SO_4^{2-} anions, TFSI^- anions are concentrated closer to the Zn surface. This is consistent with previous studies that the TFSI^- anion has a stronger hydrophobicity than SO_4^{2-} anion (Energy & Environ. Sci. 16, 1480-1501 (2023)).

In addition, we derived the energy changes of the entire system (Figure R8), which shows that the systems could be converged within 20 ps of simulation.

Furthermore, we also seek to corroborate the accuracy of the calculation results through experimental results. The experimental design is as follows. We compared the Raman signal of the bulk 1 m $\text{ZnSO}_4+0.1$ m HTFSI electrolyte with the signal at the Zn surface through surface-enhanced Raman spectroscopy. SiO_2 -coated Au nanoparticles were loaded on the Zn surface to better detect the species at the vicinity of the Zn surface. As shown in Figure R8, compared with the bulk electrolyte, the peak area ratio of $\text{TFSI}^-/\text{SO}_4^{2-}$ at the Zn surface region is higher (0.230 vs. 0.194), indicating that TFSI^- anions prefer to be enriched on the electrode surface. As a result of the strong hydrophobicity of TFSI^- , a H_2O -deficient region was formed on the Zn surface with greatly suppressed corrosion reactions.

Figure R7 Snapshots of the distribution of electrolyte species in 1 m $\text{ZnSO}_4+0.1$ m HTFSI.

Figure R8, The energy changes of the entire system within 20 ps.

Figure R9, **a** Raman spectrum of the bulk 1 m ZnSO₄+0.1 m HTFSI electrolyte and **b** Raman spectrum of the Zn surface region.

line 425-431: Surface-enhanced Raman spectroscopy (SERS) and theoretical calculations were employed to gain a better understanding of the role of HTFSI in Zn metal stability. We compared the Raman signal of the bulk 1 m ZnSO₄+0.1 m HTFSI electrolyte with the signal at the Zn surface region. As shown in Fig. 6c and 6d, compared with the bulk electrolyte, the peak area ratio of TFSI⁻/SO₄²⁻ at the Zn surface is higher (0.230 vs. 0.194), indicating that TFSI⁻ anions prefer to be enriched on the electrode surface.

line 575-579: For SERS measurements, SiO₂-shell insulated Au nanoparticles (CP-2, PERSer Nanotechnology) were dispersed on the Zn surface.

Overall, I think this is an intriguing work though it is largely inspired by Li et al

(reference #42 here). However, I think there are still some open questions about the mechanism that need answering as well as the authors should add the necessary information to replicate the simulation results. I would suggest the authors remove the AIMD simulation results to better align the remaining DFT results with the experimental observations.

REVIEWERS' COMMENTS

Reviewer #4 (Remarks to the Author):

I commend the authors for their fast turnaround time in revising the manuscript. The authors have satisfactorily addressed my concerns and provided thoughtful commentary on my questions. The simulation section now contains additional details needed to reproduce their work. It's an interesting read, I support its publication.

I will note that my concern about the double layer structure is not related to convergence of the potential energy but rather the distribution of the molecules within the EDL. Truly capturing this could take many nanoseconds as the limiting factor here is going to be ion-ion and ion-solvent exchanges, well beyond the scope of the simulations that can be performed with DFT or semi-empirical methods. The added independent replicates and use of supporting experiments however strengthen the conclusions.

To give an example of the timescales common for these types of EDL simulations, consider the details in the SI for <https://onlinelibrary.wiley.com/doi/full/10.1002/ange.202017020> in which EDL simulations were performed for 50 ns.

Response to the reviewer's comment for NCOMMS-23-39968B

Reviewer #4 (Remarks to the Author):

I commend the authors for their fast turnaround time in revising the manuscript. The authors have satisfactorily addressed my concerns and provided thoughtful commentary on my questions. The simulation section now contains additional details needed to reproduce their work. It's an interesting read, I support its publication.

Response: We are grateful for your positive comments on our work.

I will note that my concern about the double layer structure is not related to convergence of the potential energy but rather the distribution of the molecules within the EDL. Truly capturing this could take many nanoseconds as the limiting factor here is going to be ion-ion and ion-solvent exchanges, well beyond the scope of the simulations that can be performed with DFT or semi-empirical methods. The added independent replicates and use of supporting experiments however strengthen the conclusions.

To give an example of the timescales common for these types of EDL simulations, consider the details in the SI for <https://onlinelibrary.wiley.com/doi/full/10.1002/ange.202017020> in which EDL simulations were performed for 50 ns.

Response: Thank you very much for the valuable feedback and references provided. The referenced literature utilizes classical molecular dynamics (cMD) to perform electric double layer (EDL) simulations, which excels in capturing dynamics over long time scales. In addition, extending the simulation duration may be beneficial with the inclusion of an electric field. We agree with the reviewer's perspective that such cMD simulations spanning multiple nanoseconds may also capture the molecular distribution within the electric double layer (EDL).

In our case, we chose to employ the ab initio molecular dynamics (AIMD) methodology. In comparison to cMD, AIMD imposes more significant demands on computational resources concerning simulation duration. Therefore, we did not extend our simulations to nanoseconds. However, we think the chosen simulation duration is reasonable as it aligns with common standards for simulating interface-related issues in water systems using density functional theory (DFT) or semi-empirical methods (J. Chem. Phys. 149, 014707 (2018); Physical Review B82, 081406 R (2010); J. Phys. Chem. C, 114, 32 (2010)).

Once again, we appreciate the reviewer's careful guidance and suggestions, which have been instrumental in guiding our research efforts.